# Connecting Solutions and Boundary Conditions/Parameters Directly: Solving PDEs in Real Time with PINNs

## Abstract

Physics-Informed Neural Networks (PINNs) have proven to be important tools for solving both forward and inverse problems of partial differential equations (PDEs). However, PINNs face the retraining challenge in which neural networks need to be retrained once the parameters, or boundary/initial conditions change. To address this challenge, meta-learning PINNs train a meta-model across a range of PDE configurations, and the PINN models for new PDE configurations are then generated directly or fine-tuned from the meta-model. Meta-learning PINNs are confronted with either the issue of generalizing to significantly new PDE configurations or the time-consuming process of fine-tuning. By analyzing the mathematical structure of various PDEs, in this paper we establish the direct and mathematically sound connections between PDE solutions and boundary/initial conditions, sources and parameters. The learnable functions in these connections are trained offline in less than 1 hour in most cases. With these connections, the solutions for new PDE configurations can be obtained directly and vice versa, without retraining and fine-tuning at all. Our experimental results indicate that our methods are comparable to vanilla PINNs in terms of accuracy in forward problems, yet at least 400 times faster than them (even over 800 times faster for variable initial/source problems). In inverse problems, our methods are much more accurate than vanilla PINNs while being 80 times faster. Compared with meta-learning PINNs, our methods are much more accurate and about 20 times faster than fine-tuning. Our inference time is less than half a second in forward problems, and at most 3 seconds in inverse problems (less than half a second for variable initial/source problems of linear PDEs). Our code will be made publicly available upon acceptance.

## 1 Introduction

PDEs are crucial mathematical tools used to describe various phenomena in fields such as physics, chemistry, and biology. They provide precise descriptions of complex systems' dynamic behaviors and offer a theoretical foundation for system analysis, prediction, and control. In practice, PDEs are often required to be solved repetitively in forward problems under different configurations of parameters, boundary/initial conditions or sources, and it is also often necessary to repetitively find the optimal values of them in inverse problems given different constraints on solutions. Such many query type of applications includes optimal design/control, data assimilation and uncertainty quantification. Obtaining the results rapidly in each query is important for these applications. For example, it is crucial in interactive design to immediately see the PDEs solutions or optimal configurations once users change their design options.

Traditional numerical methods to solve PDEs, including finite difference and finite element methods, face inefficiencies when dealing with high-dimensional, large-scale and inverse problems. Physics-Informed Neural Networks (Raissi et al. (2019)), which utilize deep learning to solve PDEs, have gained significant attention in recent years. PINNs approximate the solutions with the predictions of neural networks, which are trained by embedding the PDE equations and boundary/initial conditions into the loss function. However, this leads to one of the fundamental limitations of PINNs: they need to be retrained when the parameters or boundary/initial conditions change, which is time-consuming

and limits their applications in many query scenarios. Current approaches to solve this retraining problem of PINNs are based on meta-learning (see section 2), in which a meta-model is trained across a range of PDE configurations and the PINNs for new PDE configurations are generated directly or fine-tuned from this meta-model. The accuracy of meta-learning PINNs is not satisfactory yet, and the fine-tuning still consumes some time and does not meet the real-time requirement.

In this paper, through in-depth investigating the mathematical structure of various PDEs, we propose mathematically sound methods to the many query problem of PINNs by establishing the direct analytic connections between PDE solutions and boundary/initial conditions, sources and parameters. The unknown parameters in these connections are learned through offline training. With these connections, the solutions for new PDE configurations can be obtained directly and vice versa, without retraining and fine-tuning at all, making the real-time inference in both forward and inverse problems practical. In contrast, vanilla and meta-learning PINNs are general but agnostic to the mathematical structure of PDEs and thus did not fully leverage the potential of PINNs. They either need time-consuming retraining or fine-tuning, or face the issue of generalizing to significantly different configurations. Also, inverse problems are largely neglected by current meta-learning PINNs researches.

We first consider linear PDEs with variable boundary/initial conditions or sources. For linear PDEs, a solution can be expressed as a linear combination of basis solutions. We train multiple PINNs offline to solve PDEs under various sine and cosine bases, thereby obtaining basis solutions. The solution corresponding to an arbitrary boundary/initial/source $g(x)$ is then obtained by the linear combination of such basis solutions using discrete Fourier transformation (DFT) of $g(x)$. This **basis solution method** is accurate and fast since no fine-tuning is required.

For PDEs with variable parameters, we directly model the solutions as polynomials of PDE parameters with learnable coefficient functions. We derive the differential equations for coefficient functions and train them offline with theoretical guarantees. With this **polynomial model**, the solutions to PDEs with new parameters can be obtained immediately and no fine-tuning is needed. We also use this polynomial model to establish the connections between solutions and variable initial conditions for nonlinear PDEs. Finally, a simpler **scaling method** is proposed for some PDEs which directly scales the solution of a canonical PINN to obtain the solutions for new parameter values.

## 2 RELATED WORK

**Physics-Informed Neural Networks.** PINNs have been successfully applied to a wide range of scientific problems, such as fluid dynamics (Rao et al. (2020); Zhu et al. (2021)), medical imaging (Sahli Costabal et al. (2020); van Herten et al. (2022)) and climate modeling (Lütjens et al. (2021)). Many works have been devoted to the training of PINNs, such as loss reweighting (Wang et al. (2021a; 2022); Yao et al. (2023); Hao et al. (2023)), resampling (Nabian et al. (2021); Zapf et al. (2022); Hanna et al. (2022); Zeng et al. (2022); Peng et al. (2022); Tang et al. (2023); Gao & Wang (2023); Lu et al. (2021); Daw et al. (2022); Lau et al. (2024)), and ill-conditioning of differential operators (Krishnapriyan et al. (2021); De Ryck et al. (2023); Rohrhofer et al. (2022); Liu et al. (2024); Rathore et al. (2024)).

Wang & Wang (2021) propose architectures that use Fourier features (Tancik et al. (2020); Ng et al. (2024)) to effectively mitigate the spectral bias of PINNs, which is not our focus in this paper.

**Many Query Problem and Meta-Learning PINNs.** The reduced basis method (RBM) (Haasdonk (2016)) is a popular numerical method for efficiently simulating parametric PDEs. It includes an offline training stage and an online stage. The offline stage selects a number of representative parameter values via a greedy algorithm and then in the online stage a rapid reduced solution is sought for each unseen parameter value. In inverse problems, numerical methods (Hasanoğlu & Romanov (2021); Isakov (2017)) usually search the unknown parameters of PDEs in an iterative manner and require to solve forward problems in each iteration, leading to high computational cost.

The conditioned PINNs method (Moseley & Markham (2020)) takes PDE parameters or boundary conditions as additional network input and trains over many different PDE configurations, allowing it to generalize without needing to be retrained. Recently, there has been increasing interest in using meta-learning to solve parametric PDEs. Representative methods include HyperPINN (de Avila Belbute-Peres et al. (2021)), MAD-PINN (Huang et al. (2022)), NRPINN (Liu et al. (2022)), Meta-

MgNet (Chen et al. (2022)) and Hyper-LR-PINNs (Cho et al. (2023)). The implementation strategies of these methods can be divided into two main types: the first type (Chen et al. (2022); de Avila Belbute-Peres et al. (2021)) involves training a meta-network to map from PDE configurations to the parameters of the main PINN network, which generally does not require fine-tuning but often necessitates multiple networks. The second type (Huang et al. (2022); Liu et al. (2022)) involves learning an effective initialization of network parameters using multiple tasks and requires fine-tuning when the PDE configuration changes, leading to higher time cost. Additionally, since meta-learning involves multi-task training, the difficulty of different tasks can affect training results. Consequently, Toloubidokhti et al. (2024) proposes the difficulty-aware task sampler (DATS), and GPT-PINN (Chen & Koohy (2024)) employs the reduced basis method for task selection. $P^2$ INNs (Cho et al. (2024)) resolve the retraining issue by modeling the solutions of parameterized PDEs via explicitly encoding a latent representation of PDE parameters.

The main differences between our methods and the above ones lie in that our methods neither require a large number of training tasks nor fine-tuning, and can solve inverse problems efficiently due to the explicitly established analytic connections between solutions and conditions/parameters.

**Operator Learning.** Operator learning is another approach to solve parametric PDEs. Representative methods include DeepONet (Lu et al. (2019)) and FNO (Li et al. (2020)), which rely on supervision from explicit solutions of different configurations to train neural networks. In comparison, our methods are unsupervised and incorporate prior knowledge of physics laws. The physics-informed DeepONet (PI-DeepONet) method (Wang et al. (2021b)) integrates physical laws into the operator learning framework to reduce the data collection burden.

## 3 PRELIMINARIES

**Physics-Informed Neural Networks.** The general form of a PDE is as follows:

$$F(u(x,t), \mu) = f, \quad x \in \Omega, \ t \in [0, T]$$
$$B(u(x,t)) = h, \quad x \in \partial\Omega, \ t \in [0, T]; \qquad I(u(x,0)) = g, \quad x \in \Omega \tag{1}$$

where $F$ is a differential operator, $B$ is an operator associated with the boundary condition and operator $I$ is for initial condition. $\Omega$ is the spatial domain and $\partial\Omega$ is its boundary, $[0, T]$ is the time domain. The functions $f$, $g$, and $h$ represent source, initial and boundary values, respectively. $\mu$ denotes the parameter of PDE. The goal of forward problems is to obtain the solution $u(x,t)$ of equation 1, while the goal of inverse problems is to find the values of $\mu$, $f$, $g$, and $h$ given the observed data $u(x_i, t_j)$ at some points $\{x_i, t_j\}$. In practice, PDEs are often required to be solved repetitively under different configurations of $\mu$, $f$, $g$, or $h$, and the optimal values of them are required to be found repetitively with different observed data $u(x_i, t_j)$.

PINNs approximate the solution $u(x,t)$ of PDEs with the prediction $u(x,t;\theta)$ of neural networks. By sampling $N_r$ collocation points from the interior domain $\mathcal{C}_r := \Omega \times (0, T)$, $N_b$ points on the boundary $\mathcal{C}_b := \partial\Omega \times [0, T]$ and $N_b$ points at the beginning $\mathcal{C}_i := \Omega$, PINNs are trained with the following loss function to enforce the PDE constraint and boundary and initial conditions,

$$L_t(\theta) = \lambda_r L_r(\theta) + \lambda_b L_b(\theta) + \lambda_i L_i(\theta), \tag{2}$$

where $L_r(\theta) = \frac{1}{N_r} \sum_{(x,t) \in \mathcal{C}_r} \|F(u(x,t), \mu) - f\|_2^2$ is the residual loss for PDEs, $L_b(\theta) = \frac{1}{N_b} \sum_{(x,t) \in \mathcal{C}_b} \|B(u(x,t)) - h\|_2^2$ is the loss for boundary conditions, and $L_i(\theta) = \frac{1}{N_i} \sum_{(x,t) \in \mathcal{C}_i} \|I(u(x,0)) - g\|_2^2$ is the loss for initial conditions. $\lambda_r$, $\lambda_b$ and $\lambda_i$ are non-negative weights assigned to different losses. When the parameters or boundary/initial/sources change, PINNs require retraining, limiting their applications in real-time scenarios.

## 4 METHODOLOGY

In this section, we will establish the direct connections between PDE solutions and boundary/initial conditions, sources or parameters. We will take the Convection, Heat, two-dimensional Poisson and Reaction equations as examples. These equations and associated boundary/initial conditions and parameter ranges are given in Table 5 in Appendix A.

## 4.1 LINEAR PDEs WITH VARIABLE BOUNDARY/INITIAL CONDITIONS OR SOURCES

For a linear PDE, if $u_i(x,t)$ is a solution, then $u(x,t) = \sum_i a_i u_i(x,t)$ is also its solution. Thus, we can generate $u_i(x,t)$ using PINNs under some known basis boundary/initial/sources, and then linearly combine them to obtain the solution $u(x,t)$ corresponding to a general boundary/initial/source $g$, where the coefficient $a_i$ comes from the spectral decomposition of $g$.

### 4.1.1 THE BASIS SOLUTION METHOD FOR VARYING INITIAL/BOUNDARY CONDITIONS

As an example, consider the Convection equation $u_t + \beta u_x = 0$ with variable initial value $g(x)$ and fixed boundary condition $u(0,t) = u(2\pi,t)$ (or other conditions, not necessarily periodic). We choose the Fourier transformation to perform spectral decomposition. The following lemma indicates that discretized $\{g(x)\}_{x=0}^{N-1}$ can be decomposed using a total of only $N+2$ sine and cosine bases.

**Lemma 1.** *A discretized arbitrary initial condition $g(x)$ $(x = 0, 1, 2 \cdots, N-1)$ can be decomposed as $g(x) = \sum_{i=0}^{N/2} a_i cos(\frac{2\pi i x}{N}) + b_i sin(\frac{2\pi i x}{N})$ using discrete Fourier transformation (DFT), where real coefficients $\{a_i, b_i\}_{i=0}^{\frac{N}{2}}$ are determined by the DFT coefficients.*

We can solve the linear PDEs to obtain $N+2$ independent solutions $\{u_i^{cos}(x,t), u_i^{sin}(x,t)\}_{i=0}^{\frac{N}{2}}$, respectively, using initial conditions $\{u_i^{cos}(x,0) = cos(\frac{2\pi i x}{N}), u_i^{sin}(x,0) = sin(\frac{2\pi i x}{N})\}_{i=0}^{\frac{N}{2}}$ and boundary conditions $\{u_i^{cos}(0,t) = u_i^{cos}(2\pi,t), u_i^{sin}(0,t) = u_i^{sin}(2\pi,t)\}_{i=0}^{\frac{N}{2}}$. Then, the solution under a general initial condition $g(x)$ is given as follows

$$u(x,t) = \sum_{i=0}^{N/2} a_i u_i^{cos}(x,t) + b_i u_i^{sin}(x,t). \tag{3}$$

The following lemma shows that such $u(x,t)$ is the desired solution.

**Lemma 2.** *$u(x,t)$ in equation 3 is the solution of linear PDEs with variable initial condition $u(x,0) = g(x)$ and the specified boundary condition.*

The proof of Lemma 1 and the value of $\{a_i, b_i\}_{i=0}^{\frac{N}{2}}$ are given in Appendix E, and the proof of Lemma 2 is given in Appendix F.

**Implementation.** Based on Lemmas 1 and 2, we train $N+2$ independent PINNs $\{\hat{u}_i^{cos}(x,t), \hat{u}_i^{sin}(x,t)\}_{u=0}^{\frac{N}{2}}$ offline to approximate the basis solutions $\{u_i^{cos}(x,t), u_i^{sin}(x,t)\}$, respectively, with the corresponding initial conditions and boundary conditions. The final solution of linear PDEs under a new initial condition $g(x)$ is given by $\hat{u}(x,t) = \sum_{i=0}^{N/2} a_i \hat{u}_i^{cos}(x,t) + b_i \hat{u}_i^{sin}(x,t)$, which can be obtained rapidly using fast Fourier transformation (FFT). A few low frequency basis solutions are enough to recover $\hat{u}(x,t)$ accurately, thereby the offline training burden can be greatly reduced.

**Inverse Problems.** Given observed data $\{\tilde{u}(x_i, t_j)\}$, the task in inverse problems is to find the optimal coefficients $\{a_i, b_i\}_{i=0}^{\frac{N}{2}}$ as follows,

$$\{a_k^\star, b_k^\star\} = argmin_{\{a_k, b_k\}} \sum_{i,j} (\sum_{k=0}^{N/2} a_k \hat{u}_k^{cos}(x_i, t_j) + b_k \hat{u}_k^{sin}(x_i, t_j) - \tilde{u}(x_i, t_j))^2. \tag{4}$$

This is a quadratic objective and can be solved accurately and rapidly using the least square method.

### 4.1.2 THE BASIS SOLUTION METHOD FOR VARYING SOURCES

We use basis solution method to solve the two-dimensional Poisson equation $\Delta u(x,y) = f(x,y)$ with variable source $f(x,y)$. We train basis solutions offline associated with Fourier basis sources, and then linearly combine basis solutions to obtain the solution corresponding to an arbitrary new source. The detail is given in Appendix B.

### 4.1.3 GENERALITY OF THE BASIS SOLUTION METHOD

Despite we take the Convection and Poisson equations with simple rectangular 2D domains and possible periodic boundary conditions as concrete examples to describe our method, our basis solution method works for general domain geometry, other types of boundary condition and high-dimensional problems. We explain this in the following.

For the boundary of a domain (possibly high-dimensional) with arbitrary geometry, the boundary values at every boundary point can be concatenated into a array $s(i) = g(\mathbf{x}_i)$, $\mathbf{x}_i \in \mathbb{R}^d$ and $\mathbf{x}_i \in \partial\Omega$, $i = 0, 1, 2, \cdots, N - 1$, and then decomposed with one-dimensional FFT as $s(i) = \sum_{k=0}^{N/2} a_k cos(\frac{2\pi ki}{N}) + b_k sin(\frac{2\pi ki}{N})$ (see Lemma 1). The one-dimensional bases $cos(\frac{2\pi ki}{N})$ and $sin(\frac{2\pi ki}{N})$ can be inverse mapped into boundary points using $g_k^{cos}(\mathbf{x}_i) := cos(\frac{2\pi ki}{N})$ and $g_k^{sin}(\mathbf{x}_i) := sin(\frac{2\pi ki}{N})$, $i = 0, 1, 2, \cdots, N - 1$, respectively, and serve as boundary conditions for the high-dimensional and general domains. Basis solutions $u_k^{cos}(\mathbf{x}, t)$ and $u_k^{sin}(\mathbf{x}, t)$ are then obtained by training PINNs with boundary conditions $g_k^{cos}(\mathbf{x})$ and $g_k^{sin}(\mathbf{x})$, respectively, for several low frequencies $k$. Given an arbitrary boundary condition $g(\mathbf{x})$, the corresponding solution is then obtained by the linear combination of basis solutions as $u(\mathbf{x}, t) = \sum_{i=0}^{N/2} a_i u_i^{cos}(\mathbf{x}, t) + b_i u_i^{sin}(\mathbf{x}, t)$. Such $u(\mathbf{x}, t)$ satisfies the linear equations under consideration, and by $u(\mathbf{x}, t) = \sum_{i=0}^{N/2} a_i g_i^{cos}(\mathbf{x}) + b_i g_i^{cos}(\mathbf{x}) = g(\mathbf{x})$, $\mathbf{x} \in \partial\Omega$, it also satisfies the Dirichlet type of boundary condition.

For the Neumann type of boundary conditions $\frac{\partial u}{\partial \mathbf{n}} = g(\mathbf{x})$, $\mathbf{x} \in \partial\Omega$, we first convert $g(\mathbf{x}_i)$, $\mathbf{x}_i \in \partial\Omega$ into an array as above, and then train basis solutions $u_k^{cos}(\mathbf{x}, t)$ and $u_k^{sin}(\mathbf{x}, t)$ with Neumann type of boundary conditions: $\frac{\partial u_k^{cos}}{\partial \mathbf{n}}(\mathbf{x}_i, t) = cos(\frac{2\pi ki}{N})$ and $\frac{\partial u_k^{sin}}{\partial \mathbf{n}}(\mathbf{x}_i, t) = sin(\frac{2\pi ki}{N})$, $\mathbf{x}_i \in \partial\Omega$, respectively. By $\frac{\partial u}{\partial \mathbf{n}}(\mathbf{x}_i, t) = \sum_{i=0}^{N/2} a_i \frac{\partial u_i^{cos}}{\partial \mathbf{n}}(\mathbf{x}_i, t) + b_i \frac{\partial u_i^{sin}}{\partial \mathbf{n}}(\mathbf{x}_i, t) = g(\mathbf{x}_i)$, $\mathbf{x}_i \in \partial\Omega$, the Neumann type of boundary condition is satisfied.

## 4.2 PDEs WITH VARIABLE PARAMETERS

### 4.2.1 THE POLYNOMIAL MODEL

In this section, we use polynomials to model the relationship between solutions and parameters. This polynomial model is inspired by the finite difference computation of solutions. We take the Convection equation as an example. The derivation for the Heat equation will be given in Appendix I. In addition to solve PDEs with variable parameters, the polynomial model is also used for the nonlinear Reaction equation with variable initial condition, as described in Appendix J.

**The Model.** We take the Convection equation $u_t + \beta u_x = 0$ as an example to describe the derivation of our polynomial model. By finite difference discretization, using $u_j^i$ to denote the approximated solution at point $(x_j, t_i)$, we have $u_j^{i+1} = (1 - \lambda\beta)u_j^i + \lambda\beta u_{j-1}^i$, where $\lambda = \frac{\tau}{h}$, $\tau$ and $h$ are time step size and spatial step size, respectively. Using this expression recursively and denoting $\gamma = \lambda\beta$, we then have $u_j^{i+2} = u_j^i(1 - 2\gamma + \gamma^2) + u_{j-1}^i(2\gamma - 2\gamma^2) + u_{j-2}^i\gamma^2$ $= u_j^i + \gamma(-2u_j^i + 2u_{j-1}^i) + \gamma^2(u_j^i - 2u_{j-1}^i + u_{j-2}^i)$, which is a polynomial of $\gamma$. By this argument, one can infer that the solution $u(x, t)$ at any point $(x, t)$ is a polynomial of $\gamma$, with the coefficients being specific to $(x, t)$ and determined by the initial value $u(x, 0)$. For a given initial condition, we can write the polynomial expression of $u(x, t)$ as $u(x, t) = \sum_{j=0}^{N_p} w_j(x, t)\gamma^j$, where the $j$th coefficient $w_j(x, t)$ is a function of space and time, $N_p$ is maximal power of $\gamma$. In finite difference method, $\gamma < 1$ is required to ensure stability, therefore for $\beta \in (0, P)$, we can write the polynomial as

$$u(x, t) = \sum_{j=0}^{N_p} w_j(x, t)(\beta/P)^j. \tag{5}$$

The remaining task is how to learn coefficient functions $w_j(x, t)$. They should make the Convection equation $u_t + \beta u_x = 0$ satisfied. Therefore,

$$\sum_{j=0}^{N_p} \partial_t w_j(x, t)(\beta/P)^j + \beta \sum_{j=0}^{N_p} \partial_x w_j(x, t)(\beta/P)^j = 0, \tag{6}$$

which leads to

$$\sum_{j=0}^{N_p} \partial_t w_j(x,t)(\beta/P)^j + P \sum_{j=1}^{N_p+1} \partial_x w_{j-1}(x,t)(\beta/P)^j = 0, \tag{7}$$

$$\sum_{j=1}^{N_p}[\partial_t w_j(x,t) + P\partial_x w_{j-1}(x,t)](\beta/P)^j + \partial_t w_0(x,t)(\beta/P)^0 + P\partial_x w_{N_p}(x,t)(\beta/P)^{N_p+1} = 0, \tag{8}$$

Since $\beta$ can have arbitrary value, we have

$$\begin{cases} \partial_t w_j(x,t) + P\partial_x w_{j-1}(x,t) = 0, & j = 1, 2, \cdots, N_p \\ \partial_t w_0(x,t) = 0, \end{cases} \tag{9}$$

$$\partial_x w_{N_p}(x,t) = 0. \tag{10}$$

We now consider the initial condition and boundary condition. The initial condition $u(x,0)) = g(x)$ yields $\sum_{j=0}^{N_p} w_j(x,0)(\beta/P)^j = g(x)$. Again by the fact that $\beta$ can be arbitrary, we have

$$\begin{cases} w_j(x,0) = 0, & j = 1, 2, \cdots, N_p \\ w_0(x,0) = g(x). \end{cases} \tag{11}$$

For the periodic boundary condition $u(0,t) = u(L,t)$, we have $\sum_{j=0}^{N_p} w_j(0,t)(\beta/P)^j = \sum_{j=0}^{N_p} w_j(L,t)(\beta/P)^j$, hence

$$w_j(0,t) = w_j(L,t), \quad j = 0, 1, \cdots, N_p. \tag{12}$$

Alternatively, if boundary condition $u(x,t) = h(x)$, $x \in \partial\Omega$ (assume $g(x) = h(x)$, $x \in \partial\Omega$) is used, by $\sum_{j=0}^{N_p} w_j(x,t)(\beta/P)^j = h(x)$, $x \in \partial\Omega$, we will have $w_j(x,t) = 0$, $j = 1, 2, \cdots, N_p$, and $w_0(x,t) = h(x)$, $x \in \partial\Omega$.

**Theoretical Analysis.** Do equations 9,10,11 and 12 have exact solutions? How accurate is the polynomial model in equation 5? We have the following theorem to answer these theoretical questions and establish the upper bound of loss for our polynomial model, whose proof is given in Appendix H.

**Theorem 1.** *For the Convection equation $u_t + \beta u_x = 0$, $x \in [0,L], t \in [0,1]$ with initial condition $u(x,0) = g(x)$ and periodic boundary condition $u(0,t) = u(L,t)$ (or generally, $u(x,t) = h(x)$, $x \in \partial\Omega$), suppose $g(x)$ is differentiable up to the $(N_p + 1)$-th order and satisfies the periodic conditions $g(0) = g(L)$ and $\frac{\partial^n g}{\partial x^n}(0) = \frac{\partial^n g}{\partial x^n}(L)$, $n = 1, 2, \cdots, N_p$ (or generally, $h(x) = g(x)$ and $\frac{\partial^n h}{\partial x^n}(x) = 0$, $n = 1, 2, \cdots, N_p$, $x \in \partial\Omega$). If we solve $w_j(x,t)$ ($j = 0, 1, 2, \cdots, N_p$) using equations 9, 11, 12 and neglecting equation 10, then $w_j(x,t)$ ($j = 1, 2, \cdots, N_p$) can be solved exactly, and the total loss $L_t = \lambda_r L_r + \lambda_b L_b + \lambda_i L_i$ is at most $\lambda_r (\max_x \frac{\partial^{N_p+1} g(x)}{\partial x^{N_p+1}})^2 (\frac{P^{N_p+1}}{N_p!}(\frac{\beta}{P})^{N_p+1})^2$.*

**Implementation.** We use neural networks to approximate the coefficient functions $w_j(x,t)$ ($j = 0, 1, \cdots, N_p$). They are offline trained using losses corresponding to equations 9,11 and 12, like in PINNs. From the loss bound given in Theorem 1, we can see that in order to control the loss, since $\frac{\beta}{P} < 1$ and the term $\frac{P^{N_p}}{N_p!}$ decreases with $N_p$ when $N_p > P$, we can increase $N_p$ to decrease the total loss. Solutions close to true counterparts will be resulted form this low loss.

In our implementation, when varying the parameter $\beta$ with fixed initial condition $g(x) = sinx$ (hence $(\max_{x\in[0,2\pi]} \frac{\partial^{N_p+1} g(x)}{\partial x^{N_p+1}})^2 = 1$) and $\lambda_r = 1$, setting $N_p = 29$ is enough to achieve $\frac{P^{N_p+1}}{N_p!} < 1$ and consequently low error for $\beta \in (0, 10]$. For analytic initial conditions, we can directly use the theoretical solutions of $w_j(x,t)$ ($j = 0, 1, \cdots, N_p$) (given in equations 31, 32 and 33 in Appendix H). If such theoretical analysis on the loss bound and the number of polynomials is unavailable for other equations, one can rely on experiments to set $N_p$.

**Inverse Problems.** Given observed data $\{\tilde{u}(x_i, t_j)\}$, the goal of inverse problems in the polynomial model is to search the optimal parameter $\beta$ based on equation 5,

$$\beta^\star = argmin_\beta \sum_{i,j} (\sum_{k=0}^{N_p} w_k(x_i, t_j)(\beta/P)^k - \tilde{u}(x_i, t_j))^2. \tag{13}$$

In our implementation, we use gradient descent optimization in PyTorch to search $\beta^\star$.

**Generality of the Polynomial Model.** Our polynomial model $u(\mathbf{x}, t) = \sum_{j=0}^{N_p} w_j(\mathbf{x}, t)(\beta/P)^j$ works for complex domains, high-dimensional problems and other types of boundary condition (Dirichlet, Neumann). The optimization of $w_j(\mathbf{x}, t)$ is similar to that of $u(\mathbf{x}, t)$ in vanilla PINNs, using residual loss and boundary/initial condition loss for $w_j(\mathbf{x}, t)$. Therefore, like vanilla PINNs, the polynomial model works for complex domains and high-dimensionality by sampling collocation points. Our polynomial model also works for both Dirichlet and Neumann boundary conditions by optimizing $w_j(\mathbf{x}, t)$ with one of them.

**Nonlinear Equations.** Our polynomial model can be extended to nonlinear equations. Take the Burgers' equation $u_t + uu_x - \nu u_{xx} = 0$ as an example. Inspired by its finite difference discretization $u_j^{n+1} = -u_j^n(1 + \frac{2\tau\nu}{h^2}) - \frac{\tau}{h}u_j^n u_j^n + \frac{\tau}{h}u_j^n u_{j+1}^n - \frac{\tau}{h^2}\nu(u_{j+1}^n - u_{j-1}^n)$, we use the polynomial expression $u(x, t) = \sum_{i=0}^{N_p} w_i(x, t)\nu^{\phi_i(x,t)}$ to model the varying parameter problem. We train $w_i(x, t)$ and $\phi_i(x, t)$ in this model in a multi-task manner using multiple values of $\nu$ with corresponding residual loss and initial/boundary condition loss for $u(x, t)$.

We can use polynomial expression $u(x, t) = \sum_{i=0}^{N_p} w_i(x, t) \prod_j (u_j^0)^{\phi_{ij}(x,t)}$ to model the varying initial condition problem. The training of it and the Navier-Stokes equation are leaved to our future work.

### 4.2.2 THE SCALING METHOD

For the Convection, Heat and Reaction equations, we can see that the derivative $u_t$ is proportional to the parameter. The scaling method is designed to deal with such equations, which is simpler and easier to implement than the polynomial model. The details of the scaling method are given in Appendix K.

## 5 EXPERIMENTS

### 5.1 EXPERIMENTAL SETUP

**Settings in Our Methods.** In this section, we experimentally verify the performance of our methods. The PDEs used in our experiments and their configurations of parameters and boundary/initial/sources are given in Table 5 in Appendix A. In our basis solution method, we set $M, N = 512$. The Convection and Heat equations are trained offline using 10 low frequency sine and cosine bases corresponding to $i = 0, 1, \ldots, 9$ in equation 3, and the Poisson equation is trained offline with 100 low frequency bases corresponding to $i, j = 0, 1, \ldots, 9$ in equation 15. In our polynomial model, the maximal power $N_p$ is set to 29 for the Convection and Heat equations and 6 for the Reaction equation, based on our theoretical analysis in Theorem 1 and Theorem 2. $N_p$ is empirically set to 40 for the Burgers' equation.

**Methods Compared.** We compare our methods with DATS (including DATS+HyperPINN and DATS+MAD-PINN) (Toloubidokhti et al. (2024)), GPT-PINN (Chen & Koohy (2024)) and vanilla PINNs using $L_2$ relative error, training and inference times as evaluation metrics. DATS+HyperPINN and GPT-PINN are only applicable to PDEs with variable parameters, and DATS+MAD-PINN is applicable to PDEs with both variable parameters and variable boundary conditions. The settings of DATS and GPT-PINN are consistent with the original papers. We also compare with PI-DeepONet (Wang et al. (2021b)) and P$^2$INNs (Cho et al. (2024)).

Our basis solution method uses bases $cos(\frac{2\pi ix}{N})$ and $sin(\frac{2\pi ix}{N})$, $i = 0, 1, \cdots, 9$ as initial conditions to train the model, therefore we use the same 20 initial conditions to train PI-DeepONet. However, PI-DeepONet requires a large number of training samples to generalize (at least 1,000 training samples of initial conditions in (Wang et al. 2021)). During testing, we use testing initial conditions (see table 6 and table 7 in Appendix D.1) that are apparently different from those used in training. Therefore, PI-DeepONet obtained a higher relative error.

**Training and Testing Tasks.** For DATS+HyperPINN and DATS+MAD-PINN, we manually specify 5 parameters in (0,10] as training tasks for the Convection and Heat equations, and 4 parameters

in (0,5] for the Reaction equation. For GPT-PINN, we only specify the same parameter ranges, and parameters used for training are selected adaptively by algorithm. For PDEs with variable boundary/initial conditions, we select a set of specific boundary/initial conditions for each equation (6 configurations for the Convection, Heat and Poisson equations, and 4 configurations for the Reaction equation). See Tables 6-12 in Appendix D for the specific configurations selected. More on training and testing tasks, and network and optimization details are given in Appendix C.

## 5.2 RESULTS

**Variable Boundary/Initial/Source Problems.** We report the $L_2$ errors of compared methods in Table 1, and offline training and online inference costs in Table 2. The reported mean errors and standard deviations are computed from the error for each instance configuration given in Tables 6-9, respectively, for each equation. For DATS+MAD-PINN, we report the training errors as in (Toloubidokhti et al. (2024)). In contrast, our methods directly generalize to arbitrary new boundary conditions without fine-tuning. It can be seen from Table 1 that the errors of our methods are close to 1% for most equations and comparable to those of vanilla PINNs in forward problems, while DATS+MAD-PINN has large errors for the considered equations due to the difficulty of simultaneous training of multiple distinct tasks. Table 2 shows that our inference time is less than half a second, on average over 800 times faster than vanilla PINNs which require retraining. Our basis solution method significantly outperforms PI-DeepOnet when training with the same set of sine and cosine initial conditions. This is due to the fact that our basis solution method accurately reconstructs arbitrary boundary/initial conditions using only a limited number of low frequency Fourier bases, while PI-DeepOnet requires large number of diverse training samples of boundary/initial conditions to generalize. Our method is also faster than PI-DeepONet in both training and inference.

Tables 1 and 2 also report the performance of our methods and vanilla PINNs on inverse problems. Our method is better than vanilla PINNs in terms of $L_2$ error. Vanilla PINN has a large error for the Poisson equation. This is due to the fact that the unknown sources at all internal collocation points need to be recovered in vanilla PINNs. In contrast, our basis solution method only needs to optimize a few coefficients associated with low frequency bases. In terms of inference time, our basis solution method usually can solve the inverse problems within half a second, on average over 1100 times faster than vanilla PINNs.

Figures 1 and 2 (and figure 6 in Appendix D.3) visualize the results of compared methods, which clearly demonstrate that our method produces satisfactory solutions and is more accurate than other methods. For the testing initial condition $u(x,0) = \sin(3x + \frac{\pi}{3})$ ($x \in [0, 2\pi]$) that has a phase shift compared with training initial conditions, the slices at t=0 and t=1 in figures 1 and 6 demonstrate that our method achieves accurate solutions (almost overlapping with the exact solutions), while the unsuccessful generalization of other methods is exhibited by the obvious shift of their solutions with respect to the exact ones.

**Variable Parameter Problems.** The mean $L_2$ errors for variable parameter problems are reported in Table 3, and offline training and online inference costs are reported in Table 4. The errors for all instance parameters are given in Tables 10-12 in Appendix D, respectively, for each equation. It can be seen from Table 3 that our polynomial and scaling methods achieve low errors that are comparable to or less than those of vanilla PINNs. The errors of DATS and GPT-PINN are much higher than ours, especially when the parameters are large as shown in Tables 10 and 12. In contrast, our polynomial and scaling methods perform consistently well for different values of parameters, showing the generalization superiority of our methods. For inverse problems, the errors of our methods are much lower than those of vanilla PINNs, due to the fact that only hundreds of sampled data points are used. The visualization in Figs.3 and 4 again shows that our methods produce much more accurate solutions than meta-learning PINNs. We then further test the extrapolation performance of the polynomial method for parameter values up to 20. We set $P = 20$ and $N_p = 60$ which is big enough to make $\frac{P^{N_p+1}}{N_p!} < 1$, and compute the solutions using equation 5 for $\beta = 1, 2, 3, \cdots, 19$ and obtain the relative errors. The results for the Convection and Heat equations are given in table 3. It is shown that our polynomial method achieves much lower errors than vanilla PINNs which encounter optimization difficulties for large parameter values (Krishnapriyan et al. (2021)). We also compare with P$^2$INNs (Cho et al. (2024)), and find that our method achieves lower errors, attributing to the explicit analytic connection between solutions and parameters in our model.

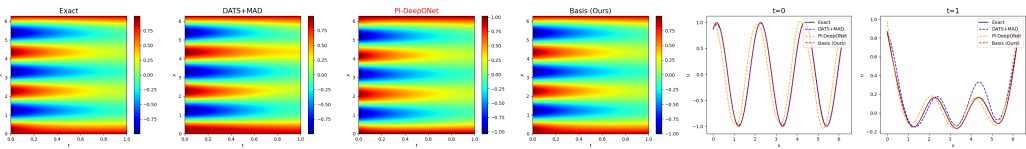

Figure 1: Prediction results of different methods for variable initial condition problem of Heat equation when $u(x, 0) = \sin(3x + \frac{\pi}{3})$.

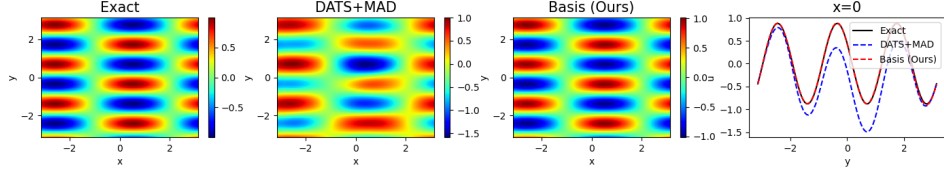

Figure 2: Prediction results of different methods for variable source problem of Poisson equation when $f(x, y) = -10\sin(x + \frac{\pi}{3})\cos(3x + \frac{\pi}{3})$.

Table 4 shows that the inference time of our methods is less than half a second, on average about 20 times faster than the fine-tuning in GPT-PINN, and over 400 times faster than vanilla PINNs. For inverse problems, our methods are over 80 times faster than vanilla PINNs which need retraining.

For the Burgers' equation with varying parameter, the results in table 3 and table 4 show that our polynomial model has achieved lower error in inverse problems and is much faster than vanilla PINN in inference (170 times faster in forward and 55 times faster in inverse), with a slightly higher error than it in foreword problems. Figure 5 visualizes the high quality prediction of our polynomial model for the Burgers' equation.

More visualizations on predictions, learned basis solutions, learned coefficient functions, canonical ans scaled solutions are provided in Appendices D.3 to D.6, respectively. Ablation studies are included in Appendix D.2 to explore the effect of the number of reserved Fourier bases and using a single network to train all basis solutions.

Table 1: The relative $L_2$ error of each method when changing the boundary/initial/source conditions.

| PDEs | Forward | | | | Inverse | |
|---|---|---|---|---|---|---|
| | Basis (Ours) | DATS+MAD | PI-DeepONet | vanilla PINN | Basis (Ours) | vanilla PINN |
| Convection | **0.014±0.006** | 0.098±0.052 | 0.534±0.053 | 0.015±0.006 | **0.014±0.006** | 0.015±0.008 |
| Heat | 0.012±0.006 | 0.098±0.023 | 0.434±0.022 | **0.003±0.003** | **0.014±0.003** | 0.025±0.016 |
| Poisson | 0.025±0.004 | 0.599±0.233 | - | **0.003±0.002** | **0.018±0.003** | 0.313±0.034 |
| Reaction | **0.009±0.001** | 0.588±0.394 | - | 0.024±0.014 | **0.001±9e-04** | 0.002±0.001 |

Table 2: Time cost of each method when changing the boundary/initial/source conditions.

| PDEs | Forward | | | | | | Inverse | |
|---|---|---|---|---|---|---|---|---|
| | Offline Training Time (h) | | | Inference Time (s) | | | Inference Time (s) | |
| | Basis (Ours) | DATS+MAD | PI-DeepONet | Basis (Ours) | PI-DeepONet | vanilla PINN (retraining time) | Basis (Ours) | vanilla PINN |
| Convection | **0.45** | 0.66 | 0.56 | **0.14** | 0.98 | 115 | **0.18** | 118 |
| Heat | **0.65** | 0.83 | 0.71 | **0.10** | 0.95 | 160 | **0.05** | 166 |
| Poisson | **5.5** | 5.78 | - | **0.40** | - | 215 | **0.41** | 193 |
| Reaction | **0.16** | 0.45 | - | **0.32** | - | 190 | **2.97** | 170 |

## 6 CONCLUSION AND FUTURE WORK

By establishing the analytic connections between PDE solutions and boundary/initial conditions, sources or parameters, we propose methods in this work to solve the retraining problem of PINNs in

which neural networks need to be retrained once the PDE configurations change. The basis solution method applies to linear PDEs with variable boundary/initial conditions or sources, the polynomial model mainly applies to linear or nonlinear PDEs with variable parameters. Our methods are very fast as well as accurate, making the applications of PINNs to interactive engineering design possible.

A limitation of our methods is that we have considered general but fixed boundary shapes, and solving PDEs with varying geometry in real-time is one of our future work. We also want to explore the problem of varying boundary/initial conditions and parameters simultaneously. Finally, we will investigate more nonlinear PDEs in our future work.

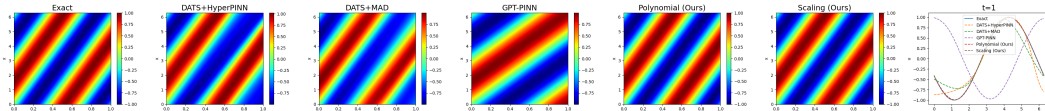

Figure 3: Prediction results of different methods for variable parameter problem of Convection equation when $\beta = 9$.

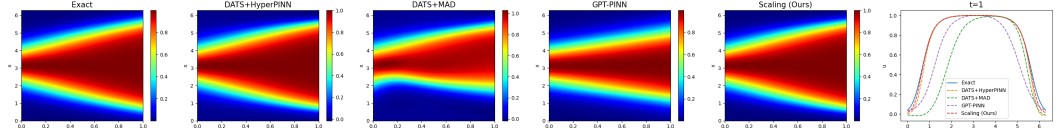

Figure 4: Prediction results of different methods for variable parameter problem of Reaction equation when $\rho = 4.8$.

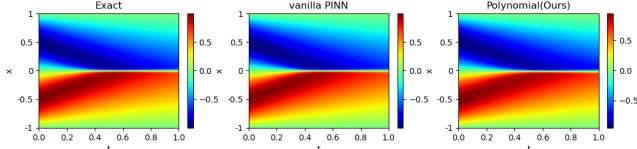

Figure 5: Prediction results of different methods for the Burgers' equation ($\nu = 0.01$).

Table 3: The relative $L_2$ error of each method when changing the parameters.

| PDE | Forward | | | | | | Inverse | |
|---|---|---|---|---|---|---|---|---|
| | Ours | DATS+Hyper | DATS+MAD | GPT-PINN | vanilla PINN | P$^2$INN | Ours | vanilla PINN |
| Convection (Polynomial) | 0.014±4e-04 | 0.108±0.071 | 0.181±0.193 | 0.128±0.214 | **0.013±5e-04** | | **0.007±0.005** | 0.489±0.470 |
| Convection (Scaling) | 0.014±7e-06 | | | | | | | |
| Heat (Polynomial) | **2e-04±4e-04** | 0.018±0.004 | 0.020±0.003 | 0.190±0.186 | 0.014±0.014 | | 0.041±0.088 | 0.112±0.139 |
| Heat (Scaling) | **0.002±4e-04** | | | | | | **2e-04±3e-04** | |
| Reaction | **0.005±0.006** | 0.011±0.009 | 0.095±0.102 | 0.056±0.089 | 0.028±0.038 | | **0.002±0.001** | 0.013±0.008 |
| Burgers (Polynomial) | 0.027±0.024 | | | | **0.011±0.005** | | **0.031±0.047** | 0.042±0.077 |
| Convection (Polynomial), $\beta \in (0, 20)$ | **0.021±0.028** | | | | 0.1978 | 0.0464 | | |
| Heat (Polynomial), $\alpha \in (0, 20)$ | **0.067±0.179** | | | | 1.2825 | 0.3745 | | |

Table 4: Time cost of each method when changing the parameters.

| PDEs | Forward | | | | | | | Inverse | |
|---|---|---|---|---|---|---|---|---|---|
| | Offline Training Time (h) | | | | Inference Time (s) | | | Inference Time (s) | |
| | Ours | DATS+Hyper | DATS+MAD | GPT-PINN | Ours | GPT-PINN | vanilla PINN (retraining time) | Ours | vanilla PINN |
| Convection (Polynomial) | **0.21(s)** | 0.78 | 0.50 | 0.27 | 0.42 | 7.2 | 156 | **2.99** | 165 |
| Convection (Scaling) | 0.13 | | | | **0.39** | | | 3.12 | |
| Heat (Polynomial) | **0.17(s)** | 0.74 | 0.67 | 0.42 | **0.41** | 6 | 164 | **3.01** | 178 |
| Heat (Scaling) | 0.04 | | | | 0.43 | | | 3.10 | |
| Reaction (Scaling) | **0.05** | 0.86 | 0.41 | 0.12 | **0.40** | 8.61 | 170 | **1.30** | 175 |
| Burgers (Polynomial) | 7.30 | | | | **1.41** | | 242 | **3.00** | 165 |

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

# Appendices

## A  EXEMPLAR PDEs AND THEIR CONFIGURATIONS

Table 5: The configurations considered for each PDE benchmark.

| PDEs | Formulations | Boundary/Initial/Source | Configurations |
|------|-------------|------------------------|----------------|
| Convection | $\frac{\partial u}{\partial t} + \beta \frac{\partial u}{\partial x} = 0$ $x \in [0, 2\pi], t \in [0, 1]$ | $u(x, 0) = sin(ax + b)$ $u(0, t) = u(2\pi, t)$ | $a \in (0, 3], b \in [0, \pi]$ $\beta \in (0, 20]$ |
| Heat | $\frac{\partial u}{\partial t} = \alpha \frac{\partial^2 u}{\partial x^2}$ $x \in [0, 2\pi], t \in [0, 1]$ | $u(x, 0) = sin(ax + b)$ $u(0, t) = u(0, 0)\ u(2\pi, t) = u(2\pi, 0)$ or $u(0, t) = u(2\pi, t)$ | $a \in (0, 3], b \in [0, \pi]$ $\alpha \in (0, 20]$ |
| Poisson | $\Delta u(x, y) = f(x, y)$ $x, y \in [-\pi, \pi]$ | $f(x, y) =$ $-(a_1^2 + b_1^2)sin(a_1 x + a_2)cos(b_1 y + b_2)$ $u(x, y)\|_{\partial\Omega} = sin(a_1 x + a_2)cos(b_1 y + b_2)\|_{\partial\Omega}$ | $a_1, b_1 \in (0, 3]$ $a_2, b_2 \in [0, \pi]$ |
| Reaction | $\frac{\partial u}{\partial t} - \rho u(1 - u) = 0$ $x \in [0, 2\pi], t \in [0, 1]$ | $u(x, 0) = \frac{\alpha h(x)}{\alpha h(x) + 1 - 0.5*h(x)}$ or $u(x, 0) = h(x)$ $h(x) = exp(-\frac{(x - \pi)^2}{2(\pi/4)^2})$ $u(0, t) = u(2\pi, t)$ | $\rho \in (0, 5]$ $\alpha \in (0, 3]$ |
| Burgers | $\frac{\partial u}{\partial t} + u\frac{\partial u}{\partial x} - \nu\frac{\partial^2 u}{\partial x^2} = 0$ $x \in [-1, 1], t \in [0, 1]$ | $u(x, 0) = -sin(\pi x)$ $u(-1, t) = u(1, t) = 0$ | $\nu \in [0.01, 0.2]$ |

## B  THE BASIS SOLUTION METHOD FOR VARYING SOURCES: THE DETAILS

Given an arbitrary discretized source $f(x, y)$ ($x = 0, 1, \cdots, M - 1; y = 0, 1, \cdots, N - 1$) and supposing $M$ and $N$ are even, we have the following decomposition of $f(x, y)$.

**Lemma 3.** $f(x, y)$ can be decomposed as

$$f(x, y) = \sum_{u=0}^{M/2} \sum_{v=0}^{N/2} [A(u, v)cos2\pi(\frac{ux}{M})cos2\pi(\frac{vy}{N}) + B(u, v)sin2\pi(\frac{ux}{M})sin2\pi(\frac{vy}{N})$$
$$+ C(u, v)cos2\pi(\frac{ux}{M})sin2\pi(\frac{vy}{N}) + D(u, v)sin2\pi(\frac{ux}{M})cos2\pi(\frac{vy}{N})], \quad (14)$$

where the four matrices $A, B, C$ and $D$ come from the two-dimensional DFT of $f(x, y)$.

The proof of Lemma 3 is presented in Appendix G. We train $4(\frac{M}{2}+1)(\frac{N}{2}+1)$ PINNs offline to obtain solutions $\{\hat{u}_{ij}^{cc}(x, y), \hat{u}_{ij}^{ss}(x, y), \hat{u}_{ij}^{cs}(x, y), \hat{u}_{ij}^{sc}(x, y)\}$, respectively, for the Poisson equation with sources $cos2\pi(\frac{ix}{M})cos2\pi(\frac{jy}{N}), sin2\pi(\frac{ix}{M})sin2\pi(\frac{jy}{N}), cos2\pi(\frac{ix}{M})sin2\pi(\frac{jy}{N}), sin2\pi(\frac{ix}{M})cos2\pi(\frac{jy}{N})$ and corresponding boundary conditions (can be defined on boundaries with arbitrary geometry). The solution for a new source $f(x, y)$ is then obtained by

$$\hat{u}(x, y) = \sum_{i=0}^{M/2} \sum_{j=0}^{N/2} A(i, j)\hat{u}_{ij}^{cc}(x, y) + B(i, j)\hat{u}_{ij}^{ss}(x, y) + C(i, j)\hat{u}_{ij}^{cs}(x, y) + D(i, j)\hat{u}_{ij}^{sc}(x, y). \quad (15)$$

The terms in equation 15 corresponding to high frequencies can be discarded without much accuracy degradation for $\hat{u}(x, y)$.

## C  MORE ON EXPERIMENTAL SETTINGS

**Training and Testing Tasks.** DATS+HyperPINN and DATS+MAD-PINN need training tasks. Since there is no fine-tuning in DATS+HyperPINN and there is no open source code for the fine-tuning in DATS+MAD-PINN, we use all selected configurations as their training tasks and no fine-tuning is used. Therefore, we only report training errors for these methods. In addition, we empirically found that when the parameters are big, gradient vanishing sometimes happens in GPT-PINN during fine-tuning. The errors before fine-tuning are thus reported for GPT-PINN. We use the same set of initial conditions as for our basis solution method, i.e., $cos(\frac{2\pi ix}{N})$ and $sin(\frac{2\pi ix}{N})$, $i = 0, 1, \cdots, 9$, to train PI-DeepONet.

In our methods, for variable boundary/initial condition problems, we fix the parameters to $\beta = 1$ for the Convection equation, $\alpha = 0.2$ for the Heat equation and $\rho = 0.1$ for the Reaction equation. For variable parameter problems, the initial conditions are fixed to $g(x) = sin(x)$ for Convection and Heat equations, and to $h(x) = exp(-\frac{(x-\pi)^2}{2(\pi/4)^2})$ for the Reaction equation. In our polynomial model for the Convection and Heat equations, we directly use our theoretical solutions of $w_j(x, t)$ $(j = 0, 1, \cdots, N_p)$ (given in equations 31, 32 and 33 in Appendix H, and equation 43 in Appendix I, respectively). The selected specific configurations in Tables 6-12 are almost all new to our methods (except $\beta = 1$ for the Convection equation and $\alpha = 1$ for the Heat equation, which are, respectively, used in the offline training of canonical equations in the scaling method). Thus, the errors reported for our methods are testing errors. For the Burgers' equation, the parameter values of $\nu$ used in training are $0.03, 0.05, 0.07, 0.09, 0.1, 0.12, 0.14, 0.16$, respectively, and those used for testing (given table 13 in Appendix D.1) are $0.01, 0.08, 0.15, 0.18, 0.20$, respectively.

**Network and Optimization.** The network architecture in DATS and GPT-PINN are all kept the same as original papers. Both our methods and vanilla PINNs are trained using fully connected neural networks of size [2, 100, 100, 100, 100, 1]. A learning rate of 1e-3 is used with the ADAM optimizer, and all methods are trained for 20,000 epochs except for GPT-PINN, whose training epochs is kept as default. A Nvidia 3090 GPU is used for the training and inference of all compared methods.

In variable boundary/initial condition problems, for the Convection, Heat and Poisson equations, our basis solution method and vanilla PINNs both sample 10,000 internal points and 100 points on each boundary. For the Reaction equation, we use 3600 internal points, 256 initial points, and 50 points on each boundary to learn $w_j(x, t)$. In our scaling method, we use 30,000 internal points for the canonical Convection equation, and 10,000 internal points for the canonical Heat and Reaction equations.

In inverse problems, 100 true values are randomly sampled for the Convection and Heat equations. Due to the large number of bases for the two-dimensional Poisson equation, 1000 points are randomly sampled. 512 points are sampled for the inverse problem of Reaction equation, and 250 points are sampled for the inverse problem of Burgers' equation. DATS and GPT-PINN did not deal with inverse problems. For vanilla PINNs, the same number of sampled data points as ours is used in inverse problems, and the number of collocation points in inverse problems is identical to that in forward problems. In contrast, our methods do not need collocation points at all in inverse problems.

$L_2$ **Error Metric for Inverse Problems.** For variable parameter problems, we directly compute the relative $L_2$ errors between optimal parameters found and their ground truth. For variable boundary/initial/source problems, the relative $L_2$ errors reported in our experiments are computed between the recovered boundary/initial/sources and their ground truth.

# D  MORE EXPERIMENTAL RESULTS

## D.1  $L_2$ ERRORS FOR EACH PDE UNDER DIFFERENT INITIAL CONDITIONS, SOURCES OR PARAMETERS

Table 6: Relative $L_2$ error for the Convection equation with variable initial condition.

| Convection | Forward | | | | Inverse | |
|---|---|---|---|---|---|---|
| Initial Condition | Basis (Ours) | DATS+MAD | PI-DeepONet | vanilla PINN | Basis (Ours) | vanilla PINN |
| $sin(x + \frac{\pi}{3})$ | 0.007 | 0.042 | 0.523 | 0.008 | 0.007 | 0.008 |
| $sin(x + \frac{2\pi}{3})$ | 0.006 | 0.045 | 0.501 | 0.007 | 0.006 | 0.007 |
| $sin(2x + \frac{\pi}{3})$ | 0.015 | 0.066 | 0.495 | 0.015 | 0.015 | 0.013 |
| $sin(2x + \frac{2\pi}{3})$ | 0.013 | 0.144 | 0.649 | 0.015 | 0.013 | 0.015 |
| $sin(3x + \frac{\pi}{3})$ | 0.023 | 0.148 | 0.507 | 0.022 | 0.023 | 0.023 |
| $sin(3x + \frac{2\pi}{3})$ | 0.021 | 0.146 | 0.532 | 0.022 | 0.021 | 0.026 |

Table 7: Relative $L_2$ error for the Heat equations with variable initial condition.

| Heat | Forward | | | | Inverse | |
|---|---|---|---|---|---|---|
| Initial Condition | Basis (Ours) | DATS+MAD | PI-DeepONet | vanilla PINN | Basis (Ours) | vanilla PINN |
| $sin(x + \frac{\pi}{3})$ | 0.007 | 0.093 | 0.405 | 0.0005 | 0.008 | 0.006 |
| $sin(x + \frac{2\pi}{3})$ | 0.005 | 0.086 | 0.453 | 0.0005 | 0.007 | 0.005 |
| $sin(2x + \frac{\pi}{3})$ | 0.013 | 0.097 | 0.463 | 0.0016 | 0.015 | 0.024 |
| $sin(2x + \frac{2\pi}{3})$ | 0.012 | 0.067 | 0.447 | 0.0016 | 0.013 | 0.035 |
| $sin(3x + \frac{\pi}{3})$ | 0.020 | 0.109 | 0.427 | 0.0026 | 0.021 | 0.041 |
| $sin(3x + \frac{2\pi}{3})$ | 0.019 | 0.135 | 0.410 | 0.0094 | 0.020 | 0.039 |

Table 8: Relative $L_2$ error for the Poisson equation with variable source.

| Poisson | Forward | | | Inverse | |
|---|---|---|---|---|---|
| Source | Basis (Ours) | DATS+MAD | vanilla PINN | Basis (Ours) | vanilla PINN |
| $-10sin(x + \frac{\pi}{3})cos(3x + \frac{\pi}{3})$ | 0.025 | 0.476 | 0.002 | 0.020 | 0.341 |
| $-10sin(x + \frac{2\pi}{3})cos(3x + \frac{2\pi}{3})$ | 0.026 | 0.353 | 0.002 | 0.022 | 0.351 |
| $-8sin(2x + \frac{\pi}{3})cos(2x + \frac{\pi}{3})$ | 0.020 | 0.654 | 0.003 | 0.014 | 0.269 |
| $-8sin(2x + \frac{2\pi}{3})cos(2x + \frac{2\pi}{3})$ | 0.020 | 0.977 | 0.003 | 0.014 | 0.273 |
| $-10sin(3x + \frac{\pi}{3})cos(x + \frac{\pi}{3})$ | 0.031 | 0.726 | 0.002 | 0.021 | 0.323 |
| $-10sin(3x + \frac{2\pi}{3})cos(x + \frac{2\pi}{3})$ | 0.030 | 0.413 | 0.007 | 0.020 | 0.323 |

Table 9: Relative $L_2$ error for the Reaction equation with variable initial condition.

| Reaction | Forward | | | Inverse | |
|---|---|---|---|---|---|
| Initial Condition | Polynomial (Ours) | DATS+MAD | vanilla PINN | Polynomial (Ours) | vanilla PINN |
| $\frac{0.1h(x)}{0.1h(x)+1-0.5h(x)}, h(x) = exp(-\frac{(x-\pi)^2}{2(\pi/4)^2})$ | 0.008 | 0.006 | 0.045 | 0.002 | 0.002 |
| $\frac{0.5h(x)}{0.5h(x)+1-0.5h(x)}, h(x) = exp(-\frac{(x-\pi)^2}{2(\pi/4)^2})$ | 0.009 | 0.696 | 0.012 | 7e-4 | 0.004 |
| $\frac{h(x)}{h(x)+1-0.5h(x)}, h(x) = exp(-\frac{(x-\pi)^2}{2(\pi/4)^2})$ | 0.009 | 0.789 | 0.020 | 5e-4 | 8e-4 |
| $\frac{3h(x)}{3h(x)+1-0.5h(x)}, h(x) = exp(-\frac{(x-\pi)^2}{2(\pi/4)^2})$ | 0.011 | 0.862 | 0.021 | 4e-4 | 6e-4 |

Table 10: Relative $L_2$ error for the Convection equation with variable parameter.

| Convection | Forward | | | | | | Inverse | | |
|---|---|---|---|---|---|---|---|---|---|
| | Polynomial | Scaling | DATS+Hyper | DATS+MAD | GPT-PINN | vanilla PINN | Polynomial | Scaling | vanilla PINN |
| $\beta = 1$ | 0.013 | 0.013 | 0.031 | 0.049 | 0.033 | 0.013 | 0.015 | 0.01 | 0.004 |
| $\beta = 3$ | 0.014 | 0.014 | 0.065 | 0.022 | 0.021 | 0.014 | 0.006 | 0.006 | 0.003 |
| $\beta = 5$ | 0.014 | 0.014 | 0.077 | 0.067 | 0.003 | 0.014 | 0.003 | 0.003 | 0.58 |
| $\beta = 7$ | 0.014 | 0.014 | 0.179 | 0.309 | 0.078 | 0.013 | 8e-04 | 0.002 | 1.02 |
| $\beta = 9$ | 0.014 | 0.015 | 0.189 | 0.460 | 0.508 | 0.013 | 0.010 | 0.001 | 0.84 |

Table 11: Relative $L_2$ error for the Heat equation with variable parameter.

| Heat | Forward | | | | | | Inverse | | |
|---|---|---|---|---|---|---|---|---|---|
| | Polynomial | Scaling | DATS+Hyper | DATS+MAD | GPT-PINN | vanilla PINN | Polynomial | Scaling | vanilla PINN |
| $\alpha = 1$ | 1e-08 | 0.001 | 0.014 | 0.019 | 0.419 | 0.002 | 4e-04 | 8e-04 | 0.002 |
| $\alpha = 3$ | 1e-07 | 0.002 | 0.018 | 0.025 | 0.031 | 0.004 | 0.002 | 9e-05 | 0.007 |
| $\alpha = 5$ | 1e-06 | 0.002 | 0.019 | 0.016 | 0.013 | 0.004 | 0.004 | 1e-04 | 0.050 |
| $\alpha = 7$ | 1e-06 | 0.002 | 0.024 | 0.020 | 0.136 | 0.027 | 8e-04 | 3e-05 | 0.170 |
| $\alpha = 9$ | 0.001 | 0.002 | 0.015 | 0.020 | 0.353 | 0.032 | 0.199 | 9e-05 | 0.330 |

Table 12: Relative $L_2$ error for the Reaction equation with variable parameter.

| Reaction | Forward | | | | | Inverse | |
|---|---|---|---|---|---|---|---|
| | Scaling (Ours) | DATS+Hyper | DATS+MAD | GPT-PINN | vanilla PINN | Scaling (Ours) | vanilla PINN |
| $\rho = 0.5$ | 0.001 | 0.007 | 0.034 | 0.008 | 0.003 | 0.004 | 0.014 |
| $\rho = 0.8$ | 0.001 | 0.004 | 0.015 | 0.004 | 0.010 | 0.003 | 0.004 |
| $\rho = 3.2$ | 0.005 | 0.008 | 0.092 | 0.024 | 0.017 | 0.001 | 0.009 |
| $\rho = 4.8$ | 0.014 | 0.025 | 0.240 | 0.189 | 0.085 | 0.002 | 0.023 |

Table 13: Relative $L_2$ error for the Burgers' equation with variable parameter.

| Burgers | Forward | | Inverse | |
|---|---|---|---|---|
| | Polynomial | vanilla PINN | Polynomial | vanilla PINN |
| $\nu = 0.01$ | 0.0731 | 0.0114 | 0.1248 | 0.1953 |
| $\nu = 0.08$ | 0.0292 | 0.0148 | 0.0054 | 0.0051 |
| $\nu = 0.15$ | 0.0118 | 0.0056 | 0.0140 | 0.0035 |
| $\nu = 0.18$ | 0.0097 | 0.0187 | 0.0023 | 0.0041 |
| $\nu = 0.2$ | 0.0090 | 0.0069 | 0.0092 | 0.0032 |

## D.2 ABLATION STUDY

### D.2.1 THE EFFECT OF NUMBER OF BASES

For our basis solution method, the number of Fourier bases is set to 10 for the Convection and Heat equations to use only lower frequency bases, since as is well-known in signal processing, the boundary/initial values primarily consist of low frequency components. We tried with more Fourier bases, including 15 and 20 bases, and the results are given in table 14, which shows that the testing errors are almost the same for different number of bases. Therefore, ten bases suffice to achieve accurate solutions.

Table 14: Relative $L_2$ testing error for the Convection equation and Heat equation with different number of bases.

| Number of Bases | Convection | | | Heat | | |
|---|---|---|---|---|---|---|
| | 10 | 15 | 20 | 10 | 15 | 20 |
| $sin(x + \frac{\pi}{3})$ | 0.0072 | 0.0072 | 0.0072 | 0.0068 | 0.0068 | 0.0068 |
| $sin(x + \frac{2\pi}{3})$ | 0.0059 | 0.0059 | 0.0059 | 0.0053 | 0.0053 | 0.0053 |
| $sin(2x + \frac{\pi}{3})$ | 0.0149 | 0.0149 | 0.0149 | 0.0138 | 0.0138 | 0.0136 |
| $sin(2x + \frac{2\pi}{3})$ | 0.0137 | 0.0138 | 0.0138 | 0.0125 | 0.0125 | 0.0125 |
| $sin(3x + \frac{\pi}{3})$ | 0.0222 | 0.0222 | 0.0222 | 0.0206 | 0.0203 | 0.0201 |
| $sin(3x + \frac{2\pi}{3})$ | 0.0212 | 0.0214 | 0.0214 | 0.0186 | 0.0186 | 0.0185 |

### D.2.2 USING A SINGLE NETWORK TO TRAIN ALL BASIS SOLUTIONS

We also train a single network to produce all basis solutions and compare with the results of training an independent network for each basis solution. The results are given in table 15. Comparing it with table 6, table 7 and table 2, one can see that training a single network yields slightly higher relative errors with smaller parameter count and faster training.

Table 15: Relative $L_2$ testing error and time cost of using a single network to train all basis solutions.

| | PDEs | Convection | Heat |
|---|---|---|---|
| Relative $L_2$ Error | $sin(x + \frac{\pi}{3})$ | 0.008 | 0.008 |
| | $sin(x + \frac{2\pi}{3})$ | 0.008 | 0.006 |
| | $sin(2x + \frac{\pi}{3})$ | 0.016 | 0.017 |
| | $sin(2x + \frac{2\pi}{3})$ | 0.015 | 0.014 |
| | $sin(3x + \frac{\pi}{3})$ | 0.023 | 0.023 |
| | $sin(3x + \frac{2\pi}{3})$ | 0.022 | 0.022 |
| Time Cost | training time (h) | 0.21 | 0.37 |
| | inference time (s) | 0.15 | 0.05 |

### D.3 VISUALIZATION OF PREDICTION RESULTS OF DIFFERENT METHODS

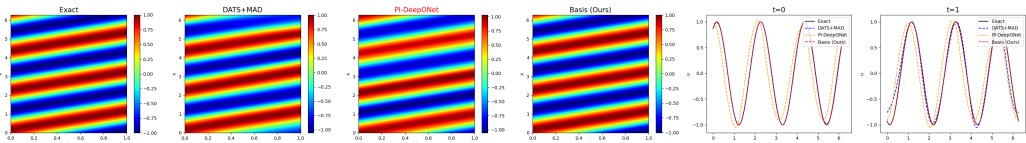

Figure 6: Prediction results of different methods for variable initial condition problem of Convection equation when $u(x,0) = \sin(3x + \frac{\pi}{3})$.

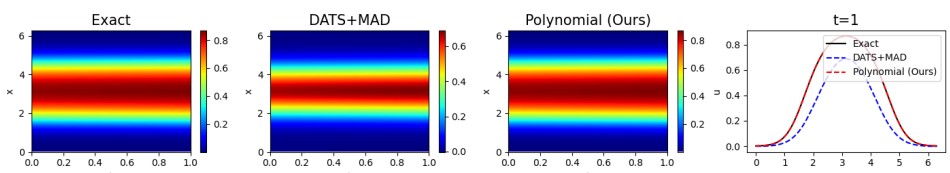

Figure 7: Prediction results of different methods for variable initial condition problem of Reaction equation when $u(x,0) = \frac{3h(x)}{3h(x)+1-0.5h(x)}$, where $h(x) = exp(-\frac{(x-\pi)^2}{2(\pi/4)^2})$.

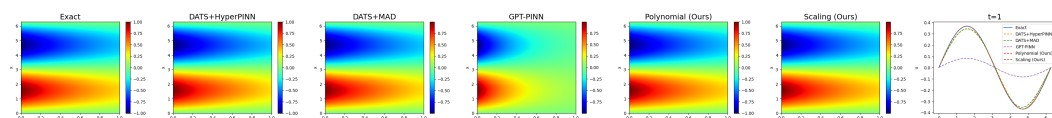

Figure 8: Prediction results of different methods for variable parameter problem of Heat equation when $\alpha$=1.

## D.4 VISUALIZATION OF LEARNED BASIS SOLUTIONS FOR THE BASIS SOLUTION METHOD

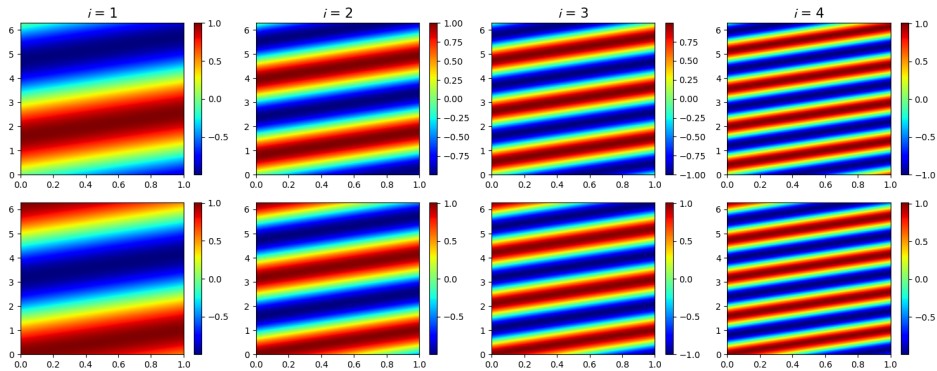

Figure 9: Visualization of basis solutions $\hat{u}_i^{sin}(x,t)$ and $\hat{u}_i^{cos}(x,t)$ ($i = 1, 2, 3, 4$) in our basis solution method: the Convection equation.

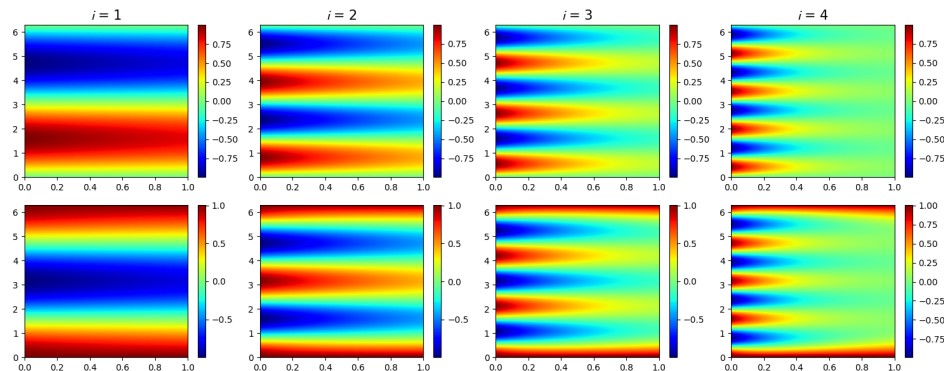

Figure 10: Visualization of basis solutions $\hat{u}_i^{sin}(x,t)$ and $\hat{u}_i^{cos}(x,t)$ ($i = 1, 2, 3, 4$) in our basis solution method: the Heat equation.

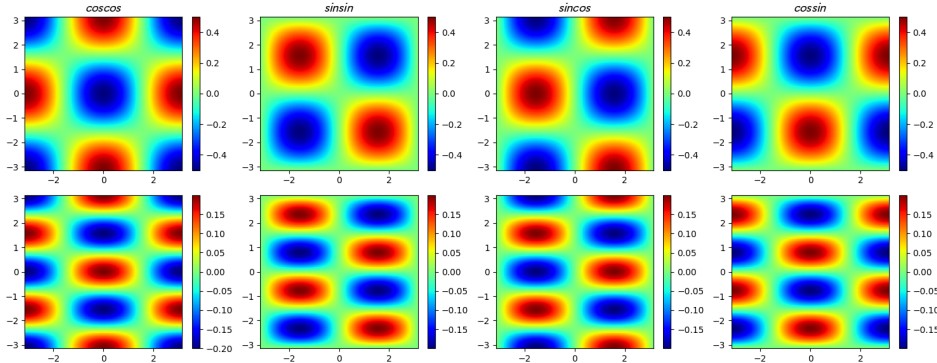

Figure 11: Visualization of basis solutions $\hat{u}_{ij}^{cc}(x,y)$, $\hat{u}_{ij}^{ss}(x,y)$, $\hat{u}_{ij}^{cs}(x,y)$ and $\hat{u}_{ij}^{sc}(x,y)$ ($i = 1$; $j = 1, 2$) in our basis solution method: the Poisson equation.

### D.5 VISUALIZATION OF LEARNED COEFFICIENT FUNCTIONS FOR THE POLYNOMIAL MODEL

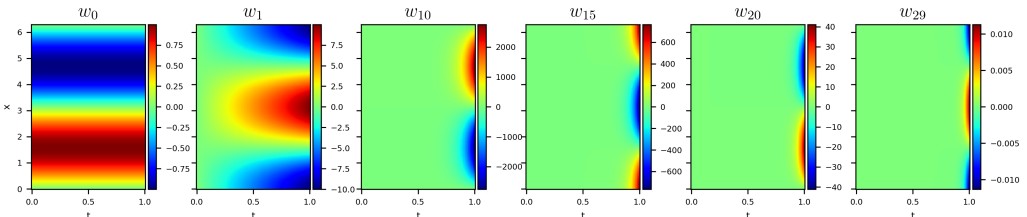

Figure 12: Visualization of learned $w_j(x, t)$ in the polynomial model of Convection equation.

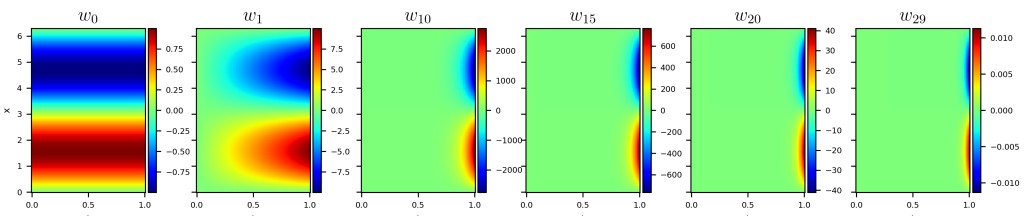

Figure 13: Visualization of learned $w_j(x, t)$ in the polynomial model of Heat equation.

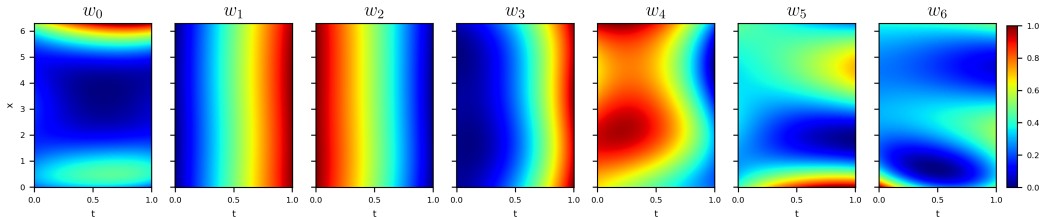

Figure 14: Visualization of learned $w_j(x, t)$ in the polynomial model of Reaction equation.

### D.6 VISUALIZATION OF CANONICAL SOLUTION AND SCALED SOLUTIONS

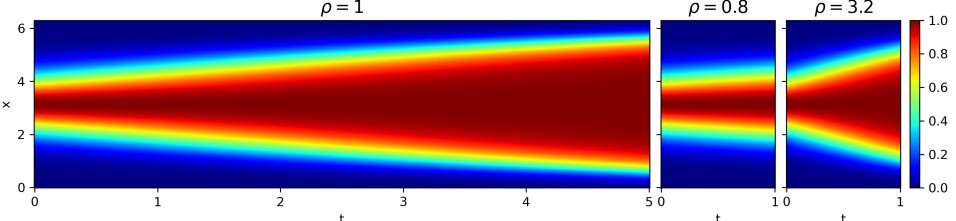

Figure 15: Visualization of canonical solution and scaled solutions for the Reaction equation.

## E PROOF OF LEMMA 1

*Proof.* Given an arbitrary initial condition $\{g(x_i)\}_{i=0}^{N-1}$ (suppose $N$ is even), its discrete Fourier transformation (DFT) and inverse discrete Fourier transformation (IDFT) are as follows, respectively,

$$ G(u) = \sum_{x=0}^{N-1} g(x)e^{-j\frac{2\pi ux}{N}}, \quad g(x) = \frac{1}{N}\sum_{u=0}^{N-1} G(u)e^{j\frac{2\pi ux}{N}}, \quad x, u = 0, 1, 2 \cdots, N-1, \quad (16) $$

where $j = \sqrt{-1}$. Let $G(u) = R(u) + jI(u)$, we have

$$g(x) = \frac{1}{N} \sum_{u=0}^{N-1} (R(u) + jI(u))(cos\frac{2\pi ux}{N} + jsin\frac{2\pi ux}{N})$$

$$= \frac{1}{N} \sum_{u=0}^{N-1} (R(u)cos\frac{2\pi ux}{N} - I(u)sin\frac{2\pi ux}{N}), \tag{17}$$

where the imaginary part in the right hand side of equation 17 is discard since $g(x)$ is real.

We now use the conjugate symmetry of DFT to reduce the number of terms in the summation, which will lead to a saving of the number of PINNs trained offline. The conjugate symmetry $G(u) = G^{\star}(N - u)$ yields $R(u) = R(N - u), I(u) = -I(N - u)$. Using $cos\frac{2\pi(N-u)x}{N} = cos(\frac{2\pi ux}{N})$ and $sin\frac{2\pi(N-u)x}{N} = -sin(\frac{2\pi ux}{N})$, we have

$$g(x) = \frac{1}{N} \sum_{u=1}^{\frac{N}{2}-1} [R(u)cos(\frac{2\pi ux}{N}) - I(u)sin(\frac{2\pi ux}{N})$$

$$+ R(N - u)cos(\frac{2\pi(N-u)x}{N}) - I(N - u)sin(\frac{2\pi(N-u)x}{N})]$$

$$+ \frac{1}{N}[R(0)cos(\frac{2\pi 0x}{N}) - I(0)sin(\frac{2\pi 0x}{N})] + \frac{1}{N}[R(\frac{N}{2})cos(\frac{2\pi\frac{N}{2}x}{N}) - I(\frac{N}{2})sin(\frac{2\pi\frac{N}{2}x}{N})]$$

$$= \frac{2}{N} \sum_{u=1}^{\frac{N}{2}-1} [R(u)cos(\frac{2\pi ux}{N}) - I(u)sin(\frac{2\pi ux}{N}] + \frac{1}{N}[R(0)cos(\frac{2\pi 0x}{N}) - I(0)sin(\frac{2\pi 0x}{N})]$$

$$+ \frac{1}{N}[R(\frac{N}{2})cos(\frac{2\pi\frac{N}{2}x}{N}) - I(\frac{N}{2})sin(\frac{2\pi\frac{N}{2}x}{N})], \quad x = 0, 1, 2 \cdots, N - 1. \tag{18}$$

Grouping the coefficients associated with different bases in equation 18 into a vector $\mathbf{a}$ and a vector $\mathbf{b}$,

$$\mathbf{a} := \left( \frac{1}{N}R(0), \left\{ \frac{2}{N}R(u) \right\}_{u=1}^{\frac{N}{2}-1}, \frac{1}{N}R(N/2) \right),$$

$$\mathbf{b} := \left( -\frac{1}{N}I(0), \left\{ -\frac{2}{N}I(u) \right\}_{u=1}^{\frac{N}{2}-1}, -\frac{1}{N}I(N/2) \right). \tag{19}$$

equation 18 can then be written as

$$g(x) = \sum_{i=0}^{N/2} a_i cos(\frac{2\pi ix}{N}) + b_i sin(\frac{2\pi ix}{N}), \quad x = 0, 1, 2 \cdots, N - 1. \tag{20}$$

Therefore, an arbitrary initial condition $\{g(x_i)\}_{i=0}^{N-1}$ can be decomposed by DFT using $N + 2$ bases $\left\{ cos(\frac{2\pi ux}{N}), sin(\frac{2\pi ux}{N}) \right\}_{u=0}^{\frac{N}{2}}$.

$\square$

## F  PROOF OF LEMMA 2

*Proof.* It is easy to see that $u(x, t)$ satisfies the linear PDEs since $u_i^{cos}(x, t)$ and $u_i^{sin}(x, t)$ satisfy them. For the initial condition, $u(x, 0) = \sum_{i=0}^{N/2} a_i u_i^{cos}(x, 0) + b_i u_i^{sin}(x, 0) = \sum_{i=0}^{N/2} a_i cos(\frac{2\pi ix}{N}) + b_i sin(\frac{2\pi ix}{N}) = g(x)$. Furthermore, by $u(0, t) = \sum_{i=0}^{N/2} a_i u_i^{cos}(0, t) + b_i u_i^{sin}(0, t) = \sum_{i=0}^{N/2} a_i u_i^{cos}(2\pi, t) + b_i u_i^{sin}(2\pi, t) = u(2\pi, t)$, the periodic boundary condition is satisfied as well. Other boundary conditions can be proved similarly. Therefore, equation 3 is the desired solution of linear PDEs under the variable initial condition. $\square$

## G    PROOF OF LEMMA 3

*Proof.* Given an arbitrary source $\{f(m,n) \mid m = 0, 1, \cdots, M-1; n = 0, 1, \cdots, N-1\}$ (suppose $M$ and $N$ are even), its two-dimensional DFT and IDFT are as follows, respectively,

$$F(u,v) = \sum_{m=0}^{M-1} \sum_{n=0}^{N-1} f(m,n) e^{-j2\pi(\frac{um}{M}+\frac{vn}{N})},$$

$$f(m,n) = \frac{1}{M \cdot N} \sum_{u=0}^{M-1} \sum_{v=0}^{N-1} F(u,v) e^{j2\pi(\frac{um}{M}+\frac{vn}{N})}. \tag{21}$$

Let $F(u,v) = R(u,v) + jI(u,v)$, we have

$$f(m,n) = \frac{1}{M \cdot N} \sum_{u=0}^{M-1} \sum_{v=0}^{N-1} [R(u,v)+jI(u,v)][cos2\pi(\frac{um}{M}+\frac{vn}{N})+jsin2\pi(\frac{um}{M}+\frac{vn}{N})]. \tag{22}$$

Using the conjugate symmetry $F(u,v) = F^*(M-u, N-v), F(0,v) = F^*(0, N-v), F(u,0) = F^*(M-u,0)$, we have $R(u,v) = R(M-u, N-v), I(u,v) = -I(M-u, N-v)$ and so on. Neglecting the imaginary part in reconstructed $f(m,n)$, we have

$$f(m,n) = \frac{1}{M \cdot N} \sum_{u=0}^{M-1} \sum_{v=0}^{N-1} [R(u,v)cos2\pi(\frac{um}{M}+\frac{vn}{N}) - I(u,v)sin2\pi(\frac{um}{M}+\frac{vn}{N})]$$

$$= \frac{1}{M \cdot N}[R(0,0)cos2\pi0 - I(0,0)sin2\pi0]$$

$$+ \frac{2}{M \cdot N} \sum_{v=1}^{\frac{N}{2}-1} [R(0,v)cos2\pi(\frac{vn}{N}) - I(0,v)sin2\pi(\frac{vn}{N})]$$

$$+ \frac{1}{M \cdot N}[R(0,\frac{N}{2})cos2\pi\frac{\frac{N}{2}n}{N} - I(0,\frac{N}{2})sin2\pi\frac{\frac{N}{2}n}{N}]$$

$$+ \frac{2}{M \cdot N} \sum_{u=1}^{\frac{M}{2}-1} [R(u,0)cos\frac{2\pi um}{M} - I(u,0)sin2\pi\frac{um}{M}] \tag{23}$$

$$+ \frac{1}{M \cdot N}[R(\frac{M}{2},0)cos2\pi\frac{\frac{M}{2}m}{M} - I(\frac{M}{2},0)sin2\pi\frac{\frac{M}{2}m}{M}]$$

$$+ \frac{2}{M \cdot N} \sum_{u=1}^{\frac{M}{2}-1} \sum_{v=1}^{N-1} [R(u,v)cos2\pi(\frac{um}{M}+\frac{vn}{N}) - I(u,v)sin2\pi(\frac{um}{M}+\frac{vn}{N})]$$

$$+ \frac{2}{M \cdot N} \sum_{v=1}^{\frac{N}{2}-1} [R(\frac{M}{2},v)cos2\pi(\frac{\frac{M}{2}m}{M}+\frac{vn}{N}) - I(\frac{M}{2},v)sin2\pi(\frac{\frac{M}{2}m}{M}+\frac{vn}{N})]$$

$$+ \frac{1}{M \cdot N}[R(\frac{M}{2},\frac{N}{2})cos2\pi(\frac{\frac{M}{2}m}{M}+\frac{\frac{N}{2}n}{N}) - I(\frac{M}{2},\frac{N}{2})sin2\pi(\frac{\frac{M}{2}m}{M}+\frac{\frac{N}{2}n}{N})].$$

For the term $\sum_{v=1}^{N-1}[R(u,v)cos2\pi(\frac{um}{M}+\frac{vn}{N})]$ in equation 23, we have

$$\sum_{v=1}^{N-1} [R(u,v)cos2\pi(\frac{um}{M}+\frac{vn}{N})]$$

$$= \sum_{v=1}^{N-1} R(u,v)[cos2\pi(\frac{um}{M})cos2\pi(\frac{vn}{N}) - sin2\pi(\frac{um}{M})sin2\pi(\frac{vn}{N})] \tag{24}$$

$$= 2 \sum_{v=1}^{\frac{N}{2}-1} R(u,v)[cos2\pi(\frac{um}{M})cos2\pi(\frac{vn}{N})]$$

$$+ R(u,\frac{N}{2})[cos2\pi(\frac{um}{M})cos2\pi(\frac{\frac{N}{2}n}{N}) - sin2\pi(\frac{um}{M})sin2\pi(\frac{\frac{N}{2}n}{N})].$$

Similarly,

$$
\begin{aligned}
&\sum_{v=1}^{N-1} [I(u,v)sin2\pi(\frac{um}{M} + \frac{vn}{N})]\\
&= \sum_{v=1}^{N-1} I(u,v)[sin2\pi(\frac{um}{M})cos2\pi(\frac{vn}{N}) + cos2\pi(\frac{um}{M})sin2\pi(\frac{vn}{N})]\\
&= 2\sum_{v=1}^{\frac{N}{2}-1} I(u,v)[sin2\pi(\frac{um}{M})cos2\pi(\frac{vn}{N})]\\
&\quad + I(u,\frac{N}{2})[sin2\pi(\frac{um}{M})cos2\pi(\frac{\frac{N}{2}n}{N}) + cos2\pi(\frac{um}{M})sin2\pi(\frac{\frac{N}{2}n}{N})].
\end{aligned}
\tag{25}
$$

Other terms in equation 23 can be expanded similarly using $cos2\pi(\frac{um}{M} + \frac{vn}{N}) = cos2\pi(\frac{um}{M})cos2\pi(\frac{vn}{N}) - sin2\pi(\frac{um}{M})sin2\pi(\frac{vn}{N})$ and $sin2\pi(\frac{um}{M} + \frac{vn}{N}) = sin2\pi(\frac{um}{M})cos2\pi(\frac{vn}{N}) + cos2\pi(\frac{um}{M})sin2\pi(\frac{vn}{N})$. Therefore, we can use

$$
\begin{aligned}
cos2\pi(\frac{ux}{M})cos2\pi(\frac{vy}{N}), &\quad sin2\pi(\frac{ux}{M})sin2\pi(\frac{vy}{N}),\\
cos2\pi(\frac{ux}{M})sin2\pi(\frac{vy}{N}), &\quad sin2\pi(\frac{ux}{M})cos2\pi(\frac{vy}{N}),\\
u = 0,1,\cdots,\frac{M}{2}; &\quad v = 0,1,\cdots,\frac{N}{2}
\end{aligned}
\tag{26}
$$

as two-dimensional DFT bases. Similar to the case of Convection equation, we group the coefficients in equation 23 associated with these bases into four matrices $A, B, C$ and $D$, and then write equation 23 as

$$
\begin{aligned}
f(x,y) = \sum_{u=0}^{M/2}\sum_{v=0}^{N/2}[&A(u,v)cos2\pi(\frac{ux}{M})cos2\pi(\frac{vy}{N}) + B(u,v)sin2\pi(\frac{ux}{M})sin2\pi(\frac{vy}{N})\\
&+ C(u,v)cos2\pi(\frac{ux}{M})sin2\pi(\frac{vy}{N}) + D(u,v)sin2\pi(\frac{ux}{M})cos2\pi(\frac{vy}{N})].
\end{aligned}
\tag{27}
$$

$\square$

## H  THE PROOF OF THEOREM 1

*Proof.* For the Convection equation, the total loss is

$$
\begin{aligned}
L_t &= \lambda_r L_r + \lambda_b L_b + \lambda_i L_i\\
&= \lambda_r \frac{1}{N_r} \sum_{(x,t)\in\mathcal{C}_r} \|u_t(x,t) + \beta u_x(x,t)\|_2^2\\
&\quad + \lambda_b \frac{1}{N_b} \sum_{t\in\mathcal{C}_b} \|u(0,t) - u(L,t)\|_2^2\\
&\quad + \lambda_i \frac{1}{N_i} \sum_{x\in\mathcal{C}_i} \|u(x,0) - g(x)\|_2^2.
\end{aligned}
\tag{28}
$$

Using the polynomial expression in 5, we have

$$
\begin{aligned}
L_t =&\lambda_r \frac{1}{N_r} \sum_{(x,t)\in\mathcal{C}_r} || \sum_{j=1}^{N_p} [\partial_t w_j(x,t) + P\partial_x w_{j-1}(x,t)](\beta/P)^j + \partial_t w_0(x,t)(\beta/P)^0 \\
&+ P\partial_x w_{N_p}(x,t)(\beta/P)^{N_p+1}||_2^2 \\
&+\lambda_b \frac{1}{N_b} \sum_{t\in\mathcal{C}_b} || \sum_{j=0}^{N_p} w_j(0,t)(\beta/P)^j - \sum_{j=0}^{N_p} w_j(L,t)(\beta/P)^j ||_2^2 \\
&+\lambda_i \frac{1}{N_i} \sum_{x\in\mathcal{C}_i} || \sum_{j=0}^{N_p} w_j(x,0)(\beta/P)^j - g(x) ||_2^2 .
\end{aligned}
\tag{29}
$$

By 9,11 and 12, we have

$$
L_t = \lambda_r \frac{P^2}{N_r} \sum_{(x,t)\in\mathcal{C}_r} \left\| \partial_x w_{N_p}(x,t)(\beta/P)^{N_p+1} \right\|_2^2 .
\tag{30}
$$

As for the solutions $w_j(x,t)$ $(j=0,1,2,\cdots,N_p)$, from 9 and 11, we have

$$
w_0(x,t) = g(x).
\tag{31}
$$

By $\partial_t w_1(x,t) = -P\partial_x w_0(x,t)$ and 11, we have

$$
w_1(x,t) = -P\frac{\partial g(x)}{\partial x}t,
\tag{32}
$$

Applying $\partial_t w_i(x,t) = -P\partial_x w_{i-1}(x,t)$ and 11 recursively and neglecting equation 10, we have

$$
w_{N_p}(x,t) = \frac{(-P)^{N_p}}{N_p!} \frac{\partial^{N_p} g(x)}{\partial x^{N_p}} t^{N_p} .
\tag{33}
$$

The periodic boundary conditions are satisfied by such $w_j(x,t)$ $(j=0,1,\cdots,N_p)$ due to $g(0) = g(L)$ and $\frac{\partial^n g}{\partial x^n}(0) = \frac{\partial^n g}{\partial x^n}(L)$, $n=1,2,\cdots,N_p$. Therefore, $w_j(x,t)$ $(j=0,1,\cdots,N_p)$ can be solved exactly if we neglect equation 10. However, since usually $\frac{\partial^{N_p+1} g(x)}{\partial x^{N_p+1}} \neq 0$, 10 may not be satisfied, thus 9, 10,11 and 12 together may have no solutions.

The total loss becomes

$$
\begin{aligned}
L_t &= \lambda_r \frac{P^2}{N_r} \sum_{(x,t)\in\mathcal{C}_r} \left\| \frac{P^{N_p}}{N_p!} \frac{\partial^{N_p+1} g(x)}{\partial x^{N_p+1}} t^{N_p} (\frac{\beta}{P})^{N_p+1} \right\|_2^2 \\
&\leq \lambda_r \frac{P^2}{N_r} \sum_{(x,t)\in\mathcal{C}_r} (\max_x \frac{\partial^{N_p+1} g(x)}{\partial x^{N_p+1}})^2 (\frac{P^{N_p}}{N_p!} (\frac{\beta}{P})^{N_p+1})^2 \\
&= \lambda_r (\max_x \frac{\partial^{N_p+1} g(x)}{\partial x^{N_p+1}})^2 (\frac{P^{N_p+1}}{N_p!} (\frac{\beta}{P})^{N_p+1})^2 .
\end{aligned}
\tag{34}
$$

This gives the upper bound of loss.

$\square$

## I THE POLYNOMIAL MODEL FOR THE HEAT EQUATION WITH VARIABLE PARAMETER

For the Heat equation $u_t = \alpha u_{xx}$ with variable parameter $\alpha \in (0, P)$, we can write the polynomial model as

$$
u(x,t) = \sum_{j=0}^{N_p} w_j(x,t)(\alpha/P)^j .
\tag{35}
$$

Substituting equation 35 into $u_t = \alpha u_{xx}$, we have

$$\sum_{j=0}^{N_p} \partial_t w_j(x,t)(\alpha/P)^j - \alpha \sum_{j=0}^{N_p} \partial_{xx} w_j(x,t)(\alpha/P)^j = 0, \tag{36}$$

which leads to

$$\sum_{j=0}^{N_p} \partial_t w_j(x,t)(\alpha/P)^j - P \sum_{j=1}^{N_p+1} \partial_{xx} w_{j-1}(x,t)(\alpha/P)^j = 0, \tag{37}$$

$$\sum_{j=1}^{N_p} [\partial_t w_j(x,t) - P\partial_{xx} w_{j-1}(x,t)](\alpha/P)^j + \partial_t w_0(x,t) - P\partial_{xx} w_{N_p}(x,t)(\alpha/P)^{N_p+1} = 0, \tag{38}$$

Since $\alpha$ can be variable, then

$$\begin{cases} \partial_t w_j(x,t) - P\partial_{xx} w_{j-1}(x,t) = 0, & j = 1, 2, \cdots, N_p \\ \partial_t w_0(x,t) = 0 \end{cases} \tag{39}$$

$$\partial_{xx} w_{N_p}(x,t) = 0. \tag{40}$$

The initial condition $u(x,0) = g(x)$ yields $\sum_{j=0}^{N_p} w_j(x,0)(\alpha/P)^j = g(x)$, thus

$$\begin{cases} w_j(x,0) = 0, & j = 1, 2, \cdots, N_p \\ w_0(x,0) = g(x) \end{cases} \tag{41}$$

For the periodic boundary condition $u(0,t) = u(L,t)$, we have

$$w_j(0,t) = w_j(L,t), \quad j = 0, 1, \cdots, N_p. \tag{42}$$

We have the following theorem to establish the bound of loss of our polynomial model for the Heat equation.

**Theorem 2.** *For the Heat equation $u_t = \alpha u_{xx}$, $x \in [0,L], t \in [0,1]$ with initial condition $u(x,0) = g(x)$ and periodic boundary condition $u(0,t) = u(L,t)$, suppose $g(x)$ is differentiable up to the $(2N_p + 2)$-th order and satisfies the periodic conditions $g(0) = g(L)$ and $\frac{\partial^n g}{\partial x^n}(0) = \frac{\partial^n g}{\partial x^n}(L)$, $n = 2, 4, \cdots, 2N_p$. If we solve $w_j(x,t)$ $(j = 0, 1, 2, \cdots, N_p)$ using equations 39, 41 and 42 and neglect equation 40, then $w_j(x,t)$ $(j = 0, 1, 2, \cdots, N_p)$ can be solved exactly, and the total loss $L_t = \lambda_r L_r + \lambda_b L_b + \lambda_i L_i$ is at most $\lambda_r(\max_x \frac{\partial^{2N_p+2} g(x)}{\partial x^{2N_p+2}})^2 (\frac{P^{N_p+1}}{N_p!}(\frac{\alpha}{P})^{N_p+1})^2$.*

*Proof.* Applying equation 39 and equation 41 recursively and neglecting equation 40, we have

$$w_0(x,t) = g(x),$$

$$w_1(x,t) = P\frac{\partial^2 g(x)}{\partial x^2} t, \cdots,$$

$$w_{N_p}(x,t) = \frac{P^{N_p}}{N_p!} \frac{\partial^{2N_p} g(x)}{\partial x^{2N_p}} t^{N_p}. \tag{43}$$

The periodic boundary conditions are satisfied by $w_j(x,t)$ $(j = 0, 1, \cdots, N_p)$ due to $g(0) = g(L)$ and $\frac{\partial^n g}{\partial x^n}(0) = \frac{\partial^n g}{\partial x^n}(L)$, $n = 2, 4, \cdots, 2N_p$. Therefore, $w_j(x,t)$ $(j = 0, 1, \cdots, N_p)$ can be solved exactly if we neglect equation 40. The total loss is

$$L_t = \lambda_r \frac{P^2}{N_r} \sum_{(x,t)\in\mathcal{C}_r} \left\| \frac{P^{N_p}}{N_p!} \frac{\partial^{2N_p+2} g(x)}{\partial x^{2N_p+2}} t^{N_p} (\frac{\alpha}{P})^{N_p+1} \right\|_2^2$$

$$\leq \lambda_r (\max_x \frac{\partial^{2N_p+2} g(x)}{\partial x^{2N_p+2}})^2 (\frac{P^{N_p+1}}{N_p!}(\frac{\alpha}{P})^{N_p+1})^2. \tag{44}$$

This completes the proof.

$\square$

### I.1 IMPLEMENTATION

When varying the parameter $\alpha$ with fixed initial condition $g(x) = sinx$ and $\lambda_r = 1$, we set $N_p = 29$ in our experiments and achieve very low error for $\alpha \in (0, 10]$.

## J THE POLYNOMIAL MODEL FOR THE REACTION EQUATION WITH VARIABLE INITIAL CONDITION

The Reaction equation $u_t - \rho u(1-u) = 0$ is a nonlinear ordinary differential equation. We assume the parameter $\rho$ is fixed and only consider to vary the initial condition. From the finite difference discretization $u_j^{i+1} = u_j^i + \tau \rho u_j^i(1 - u_j^i) = u_j^i(1 + \tau\rho) - (u_j^i)^2\tau\rho$, we can infer that the solution $u_j^i$ is a polynomial of initial value $u_j^0$, thus we model the relationship between the solution $u(x, t)$ and initial condition $g(x)$ as follows,

$$u(x, t) = \sum_{j=0}^{N_p} w_j(x, t)g^j(x). \tag{45}$$

Substituting equation 45 into $u_t - \rho u(1-u) = 0$, we have

$$\sum_{j=0}^{N_p} \partial_t w_j(x,t)g^j(x) - \rho\sum_{j=0}^{N_p} w_j(x,t)g^j(x)(1 - \sum_{k=0}^{N_p} w_k(x,t)g^k(x)) = 0, \tag{46}$$

which leads to

$$\sum_{j=0}^{N_p} \partial_t w_j g^j - \rho\sum_{j=0}^{N_p} w_j g^j + \rho\sum_{j,k=0}^{N_p} w_j w_k g^{j+k} = 0. \tag{47}$$

Since $g(x)$ can be arbitrary, we have

$$\begin{cases} \partial_t w_i - \rho w_i + \rho\sum_{\{j,k=0,1,\cdots,N_p|j+k=i\}} w_j w_k = 0, & i = 0, 1, 2, \cdots, N_p, \\ \sum_{\{j,k=1,2,\cdots,N_p|j+k=i\}} w_j w_k = 0, & i = N_p + 1, N_p + 2, \cdots, 2N_p, \end{cases} \tag{48}$$

The initial condition $u(x, 0) = g(x)$ leads to

$$\begin{cases} w_j(x, 0) = 0, & j = 0, 2, 3, \cdots, N_p \\ w_1(x, 0) = 1. \end{cases} \tag{49}$$

The periodic boundary condition $u(0, t) = u(L, t)$ leads to

$$w_j(0, t) = w_j(L, t), \quad j = 0, 1, \cdots, N_p. \tag{50}$$

We then train neural networks to approximate the coefficient functions $w_j(x, t)$ $(j = 0, 1, \cdots, N_p)$ using losses associated with equations 48, 49 and 50.

For inverse problems, based on equation 45, we use gradient descent search to find the initial values $g(x_i)$ at discretized points $\{x_i\}$.

In our implementation, the analytic solution to the Reaction equation is given by $u(x, t) = \frac{\alpha h(x)e^{\rho t}}{\alpha h(x)e^{\rho t}+1-0.5h(x)}$, where $h(x) := exp(-\frac{(x-\pi)^2}{2(\pi/4)^2})$. Therefore, the initial condition is $u(x, 0) = \frac{\alpha h(x)}{\alpha h(x)+1-0.5h(x)}$. We vary the value of $\alpha$ to change the initial condition. Note that $0 < 1-0.5h(x) < 1$ since $0 < h(x) \leq 1$, we have $g(x) = u(x, 0) < 1$ if $\alpha > 0$. Thus, the term $g^j(x)$ in equation 45 decreases exponentially with $j$, and we found in our experiments that $N_p = 6$ is enough to achieve low approximation error.

## K THE DETAILS OF THE SCALING METHOD

### K.1 CANONICAL SOLUTION AND SCALED SOLUTIONS

Take the Convection equation as an example to describe our scaling method, which is simpler and easier to implement than the polynomial model. Suppose the boundary/initial conditions are fixed.

We call the Convection equation $u_t + u_x = 0$ (with $\beta = 1$) as the canonical Convection equation, and train a PINN to approximate its solution $u(x, t)$. Given a Convection equation $u_t + \beta u_x = 0$ ($\beta \neq 1$) with unknown solution $u_\beta(x, t)$, we want to scale $u(x, t)$ to obtain $u_\beta(x, t)$. We have the following lemma to achieve this goal, whose proof in provided in Appendix K.2.

**Lemma 4.** *The function $u_\beta(x, t) := u(x, \beta t)$ is the solution of the equation $\frac{\partial u_\beta(x,t)}{\partial t} + \beta \frac{\partial u_\beta(x,t)}{\partial x} = 0$ ($\beta \neq 1$) with initial condition $u_\beta(x, 0) = g(x)$ and periodic boundary condition $u_\beta(0, t) = u_\beta(L, t)$ (or other conditions, not necessarily periodic), where $u(x, t)$ is the solution of canonical Convection equation with initial condition $u(x, 0) = g(x)$ and boundary condition $u(0, t) = u(L, t)$ (or other non-periodic boundary conditions).*

**Implementation.** When training PINNs to approximate the canonical solution $u(x, t)$, for $\beta \in (0, P]$, the scaled time domain $[0, PT]$ is used, which will require more collocation points if $P \gg 1$. We then scale the PINNs' canonical solutions $\hat{u}(x, t)$ to obtain $u_\beta(x, t) = \hat{u}(x, \beta t)$.

### K.2 PROOF OF LEMMA 4

*Proof.* By canonical equation, we have

$$\frac{\partial u(x', t')}{\partial t'} + \frac{\partial u(x', t')}{\partial x'} = 0. \tag{51}$$

Let $x' = x, t' = \beta t$, we then have

$$\frac{\partial u(x, \beta t)}{\partial t} \frac{\partial t}{\partial t'} + \frac{\partial u(x, \beta t)}{\partial x} \frac{\partial x}{\partial x'} = \frac{\partial u(x, \beta t)}{\beta \partial t} + \frac{\partial u(x, \beta t)}{\partial x} = 0. \tag{52}$$

Therefore,

$$\frac{\partial u_\beta(x, t)}{\partial t} + \beta \frac{\partial u_\beta(x, t)}{\partial x} = 0. \tag{53}$$

By

$$u_\beta(x, 0) = u(x, \beta 0) = g(x), \quad u_\beta(0, t) = u(0, \beta t) = u(L, \beta t) = u_\beta(L, t), \tag{54}$$

the initial condition $u_\beta(x, 0) = g(x)$ and boundary condition $u_\beta(0, t) = u_\beta(L, t)$ are also satisfied by $u_\beta(x, t)$. Consequently, $u_\beta(x, t) := u(x, \beta t)$ is the desired solution. $\square$

### K.3 INVERSE PROBLEMS

Given observed data $\{\tilde{u}(x_i, t_j)\}$, the goal of inverse problems in the scaling method is to obtain the optimal parameter $\beta$. This is achieved by the following optimization problem,

$$\beta^\star = argmin_\beta \sum_{i,j} (\hat{u}(x_i, \beta t_j) - \tilde{u}(x_i, t_j))^2. \tag{55}$$

In our implementation, we use gradient descent optimization in PyTorch to search $\beta^\star$, in which the gradient of $\hat{u}(x_i, \beta t_j)$ with respect to $\beta$ is fulfilled by the auto-differentiation since $\hat{u}(x_i, \beta t_j)$ is the output of a neural network.

