# OpenReview forum: "Connecting Solutions and Boundary Conditions/Parameters Directly: Solving PDEs in Real Time with PINNs"
_ICLR.cc/2025/Conference — Submitted to ICLR 2025_

### Official Review · Reviewer_WPj4 · 2024-10-20

**Soundness:** 2
**Presentation:** 2
**Contribution:** 2
**Rating:** 3
**Confidence:** 4

**Summary:**

This paper addresses one limitation of PINNs in solving PDEs: the need for retraining when parameters or boundary/initial conditions change. The authors propose three new methods to directly connect PDE solutions with their configurations: a basis solution method for linear PDEs, a polynomial model for variable parameters, and a scaling method for certain PDEs. These approaches enable real-time inference without retraining, potentially allowing for faster, more efficient PDE solving in many-query scenarios. The methods are tested on several PDEs. Results indicate comparable accuracy to traditional PINNs but with significantly faster inference times, and improved performance over meta-learning PINNs. Theoretical analysis is provided to establish error bounds.  While the work shows some promises, its generalizability to more complex PDEs and real-world scenarios may require further investigation.

**Strengths:**

1. The proposed approaches are vaildated by several examples, and the results are promising  in terms of speed and accuracy compared to existing methods.

2. The authors provide some theoretical analysis to support their methods.

**Weaknesses:**

1. A significant limitation of the proposed methods is their strong dependence on the specific structure of the PDEs and their associated boundary and initial conditions. In these cases, the solution spaces are effectively reduced to finite-dimensional spaces, where all solutions can be expressed as simple transformations of basis functions within the solution space. This approach, while effective for the presented examples, raises questions about its generalizability. It remains unclear how these methods could be extended, even in principle, to more complex PDEs or realistic applications that may not conform to such simplified structures.

2. The paper lacks a crucial comparison with physics-informed neural operators, such as PI-DeepONet and PI-FNO. Given that the primary goal of this work is to address the limitations of PINNs in solving parametric PDEs, these physics-informed neural operator methods are highly relevant as they can also potentially address the same issues with PINNs that the authors are targeting. The absence of comparisons against these important baselines leaves a significant gap in the evaluation of the proposed methods' effectiveness and novelty relative to the current state-of-the-art in the field.

**Questions:**

1. Given the limitations discussed, could the authors elaborate on potential strategies to generalize their methods to more complex PDEs? Specifically:
    * For a general linear PDE system with non-zero initial and boundary conditions, how might one decompose the solution into a sum of basis solutions?
   * Consider a simple nonlinear PDE like the Burgers' equation, u_t + u u_x - ν u_xx = 0. How could the proposed methods be applied to learn the mapping from initial conditions to solutions at a given time t?


2. Based on the numerical problem setups presented in the paper, it appears that solving a finite number of PDEs would suffice. In this context, what advantages do the proposed methods offer over direct numerical solvers? Unlike typical PINN method papers that may provide novel insights in PINNs theory and training, this work seems to construct specific PDE problems and solve them using PINNs as a tool. Could the authors clarify the benefits of their approach compared to conventional numerical methods in these scenarios?

3. For Table 2, could the authors provide clarification on the reported inference time for vanilla PINNs? Does this represent the inference time after training the PINN models, or does it include the training time? If it denotes only the inference time, the reported values seem unexpectedly slow for what should be a straightforward model evaluation.

---

> ### Author Response · Authors · 2024-11-21
> **Response to reviewer WPj4**
>
> Thank you very much for your insightful comments and valuable suggestions.
>
> 1.
> Q: strong dependence on the specific structure of the PDEs and their associated boundary and initial conditions.
>
> A: Despite we take the Convection and Poisson equations etc. with simple rectangular domains and possible periodic boundary condition as concrete examples, our basis solution method and polynomial method actually work for general domain geometry, general boundary/initial conditions and high-dimensional problems. Please see the answer to question 1 in our responses to common questions for the detailed explanation.
>
> Our Fourier basis solution method applies to any linear PDEs,  and the polynomial model is generally applicable to both linear and nonlinear equations. Please also see the answers to question 2 and question 3 in our responses to common questions for more explanations.
>
> 2.
> Q: a crucial comparison with physics-informed neural operators, such as PI-DeepONet and PI-FNO.
>
> A: Thank you very much for your suggestions. We performed experiments to compare our methods with PI-DeepONet (Wang et al. 2021). The results are shown in table 1 and table 2 in the updated manuscript, which show that our basis solution method significantly outperforms PI-DeepONet when training with the same set of input initial conditions. This is because our Fourier basis solution method provides accurate reconstruction of arbitrary boundary/initial conditions using only a limited number of Fourier bases, while PI-DeepONet requires a large number of diverse training samples of boundary/initial conditions to generalize. Our method is also faster than PI-DeepONet in both training and inference. Furthermore, inverse problem is not supported by PI-DeepONet.
>
> 3.
> Q: For a general linear PDE system with non-zero initial and boundary conditions, how might one decompose the solution into a sum of basis solutions?
>
> A: The decomposition of a solution into a sum of basis solutions is given by equation 3 in our manuscript, i.e., $u(x,t) = \sum\^{N/2}\_{i=0} a\_i u\^{cos}\_i(x,t) + b\_i u\^{sin}\_i(x,t)$, where $u\^{cos}\_i(x,t)$ and $u\^{sin}\_i(x,t)$ are basis solutions, $a\_i$ and $b\_i$ are derived from the Fourier decomposition of the initial/boundary condition under consideration. This decomposition applies to general linear PDEs with arbitrary domain and arbitrary initial/boundary conditions. Please see the answer to question 1 in our responses to common questions for the detailed explanation.
>
> 4.
> Q: Consider a simple nonlinear PDE like the Burgers' equation, $u\_t + u u\_x - \nu u\_xx = 0$. How could the proposed methods be applied to learn the mapping from initial conditions to solutions at a given time t?
>
> A: Thank you very much for your questions. Please see the answer to question 2 in our responses to common questions for the detailed explanation of extending the polynomial model to the Burgers' equation.  We have added new experimental results for the Burgers' equation in the updated manuscript.
>
> 5.
> Q: what advantages do the proposed methods offer over direct numerical solvers?
>
> A: Our methods are variants of vanilla PINNs to resolve the retraining problem of them. The training of basis solutions and coefficient functions is still similar to that of vanilla PINNs. Therefore, our methods share the same advantages offered by vanilla PINNs over direct numerical solvers, such as mesh-free, continuous prediction, applicable to arbitrary domain and high-dimensional problems, efficient for inverse problems and capable of incorporating noisy sensor data.
>
> 6.
> Q: clarification on the reported inference time for vanilla PINNs.
>
> A: Thank you very much for pointing out this. Here the inference time of vanilla PINNs is the retraining time of them, since retraining is required in vanilla PINNs for each testing case. We add explanations in table 2 an table 4 to avoid confusion.
>
> Thank you again for your insightful comments and constructive suggestions which help us to improve our manuscript.
>
> [1] Sifan Wang, Hanwen Wang, and Paris Perdikaris. Learning the solution operator of parametric partial differential equations with physics-informed deeponets. Science advances, 7(40), 2021.

---

> > ### Comment · Reviewer_WPj4 · 2024-11-24
> >
> > thank the authors for their detailed response. Unfortunately, their reply does not fully address my concerns.
> >
> > Regarding my first question (Q1), it remains unclear how the proposed method can handle varying initial conditions. The newly added numerical experiments consider a fixed initial condition with changing viscosity, which does not align with the scenario I raised.
> >
> > For the results of PI-DeepONet, I am not convinced by the reported performance. The error exceeding 40% for a relatively simple problem is unexpected and inconsistent with my prior experience working with this framework. Additionally, this result appears to conflict with Figure 1, which shows PI-DeepONet performing comparably to other methods, at least for the plotted  data sample. This discrepancy raises further doubts about the reported error values. Could the authors clarify this apparent inconsistency?
> >
> > I suggest the authors conduct a comparison for Burgers' equation using exactly the same setup as in PI-DeepONet to provide a fair and consistent benchmark.
> >
> > In conclusion, the method presented in this work seems limited, and the numerical results lack sufficient robustness and clarity. Consequently, I have to maintain my original score.

---

> > > ### Author Response · Authors · 2024-11-25
> > > **response to Official Comment by Reviewer WPj4**
> > >
> > > Thank you very much for your further feedback and discussions! In our experiments, the comparison with PI-DeepONet for the varying initial condition problems is performed on the Convection and Heat equations, and the relative error results are given in table 1. The visualization results for the Convection and Heat equations are given in figure 1 and figure 6 (in Appendix D.3), respectively.
> > >
> > > **The reason why PI-DeepONet performs poorly in our experiments is as follows**. Our basis solution method uses bases $ cos(\frac{2\pi ix}{N}) $ and $sin(\frac{2\pi ix}{N})$, $i=0,1,\cdots, 9$ as initial conditions to train the model, therefore we use the same 20 initial conditions to train PI-DeepONet. However, PI-DeepONet requires a large number of training samples to generalize (at least 1,000 training samples of initial conditions in (Wang et al. 2021)). During testing, we use testing initial conditions (see table 6 and table 7 in Appendix D.1) that are apparently different from those used in training. Therefore, PI-DeepONet obtained a higher relative error.
> > >
> > > For the visualization results in figure 1 and figure 6 (in Appendix D.3), we use the testing initial condition $u(x,0)=\sin(3x+\frac{\pi}{3})$, which is significantly different than $ cos(\frac{2\pi ix}{N}) $ and $sin(\frac{2\pi ix}{N})$ used in training and is thus hard for PI-DeepONet to generalize. Though the 3rd panel for PI-DeepONet in figure 1 looks good at first glance, **it actually has a translation relative to the exact solution.** This is more apparent in the last panel (the slice at time t=1) in figure 1 which shows that the solution of PI-DeepONet (the yellow curve) is shifted towards left (less x) relative to the exact solution and our solution. To show this more clearly, **we add a new panel (the 5th panel, showing the slice at time t=0) in which the shift effect is more manifest**. Please also see figure 6 for the shift effect. Therefore, the relative error of PI-DeepONet is indeed much higher than our method. Moreover, the inverse problems are not supported by PI-DeepONet.
> > >
> > > In contrast, our basis solution method for the varying initial condition problem only needs 20 sine and cosine initial conditions to train and then accurately generalizes to arbitrary new initial conditions. This is enabled by the accurate reconstruction property of Fourier transform. **How and why our method can handle the varying initial condition problem are explained as follows.** We train basis solutions $u\^{cos}\_i(x,t)$ and $u\^{sin}\_i(x,t)$, $i=0,1,\cdots,9$ using initial conditions $u\^{cos}\_i(x,0)=cos(\frac{2\pi ix}{N})$ and $u\^{sin}\_i(x,0)=sin(\frac{2\pi ix}{N})$, $i=0,1,\cdots,9$, respectively.  Given a general initial condition $g(x)$, we first perform Fourier decomposition $g(x) \approx \sum\^{9}\_{i=0} a\_i cos(\frac{2\pi ix}{N}) + b\_i sin(\frac{2\pi ix}{N})$, and then the solution for initial condition $g(x)$ is given by $u(x,t) = \sum\^{9}\_{i=0} a\_i u\^{cos}\_i(x,t) + b\_i u\^{sin}\_i(x,t)$. This is because such $u(x,t)$ satisfies the linear equations under consideration (since $u\^{cos}\_i(x,t)$ and $u\^{sin}\_i(x,t)$ satisfy),  and by $u(x,0) = \sum\^{9}\_{i=0} a\_i u\^{cos}\_i(x,0) + b\_i u\^{sin}\_i(x,0) = \sum\^{9}\_{i=0} a\_i cos(\frac{2\pi ix}{N}) + b\_i sin(\frac{2\pi ix}{N}) \approx g(x)$, it also satisfies the initial condition. For more detailed explanations, please see our proof of Lemma 2 in Appendix F and the paragraphs in section 4.1.3.
> > >
> > > The new experiment on the Burgers' equation is for the varying parameter problem, so the initial condition is fixed and only the viscosity is changed.
> > >
> > > Thank you again for your feedback!
> > >
> > > [1] Sifan Wang, Hanwen Wang, and Paris Perdikaris. Learning the solution operator of parametric partial differential equations with physics-informed deeponets. Science advances, 7(40), 2021.

---

> > > > ### Comment · Reviewer_WPj4 · 2024-11-27
> > > >
> > > > Thank you to the authors for their further clarification. However, I believe the authors have overlooked the nonlinearity of the Burgers' equation, and as a result, the derivation appears to be incorrect. Specifically, for Burgers' equation, the proposed solution $u(x, t)=$ $\sum_{i=0}^9 a_i u_i^{\cos }(x, t)+b_i u_i^{\sin }(x, t)$ does not satisfy the PDE due to the presence of the nonlinear term $u u_x$.
> > > >
> > > > Mathematically, for nonlinear PDEs, even if $u_1$ and $u_2$ individually satisfy the equation, their sum $u_1+u_2$ does not generally satisfy the PDE.
> > > >
> > > > Additionally, when considering nonzero boundary conditions, such as $u(t,0) = u(t, 1)=1$, this method would fail or need more modifications. So, I must maintain my original score.

---

> ### Author Response · Authors · 2024-11-27
> **Response to Official Comment by Reviewer WPj4**
>
> Thank you very much for your further feedback! We make the following clarifications. Our basis solution method, in the form of $u(x, t)=$ $\sum_{i=0}^9 a_i u_i^{\cos }(x, t)+b_i u_i^{\sin }(x, t)$, is for linear equations. We did not use our basis solution method to the Burgers' equation. We only used it for linear equations like Convection and Heat with varying initial/boundary conditions and compared it with PI-DeepONet etc. (in table 1 and table 2). We used our polynomial model for the Burgers' equation with varing parameter (in table 3 and table 4). These are two different kinds of problems.  We have clearly stated the scenarios in which our basis solution method and polynomial model are used, respectively, in section 4, experimental results and conclusion in  the
> previous version of our manuscript. So, this is just a misunderstanding, and I apologize for it.
>
> When considering nonzero boundary conditions, such as $u(t,0) = u(t, 1)=1$, our basis solution method is still applicable by using $u(t,0) = u(t, 1)=1$ as boundary condition when training PINNs to learn basis solutions for varing initial conditions. We have pointed out this in the previous version of our manuscript in line 171 and Lemma 2 (both marked in red color).
>
> Thank you again for your comments. If you have any additional feedback, we would be most grateful to receive it.

---

> > ### Author Response · Authors · 2024-12-03
> > **Looking forward to your feedback**
> >
> > Dear reviewer WPj4,
> >
> > Thank you very much for your time and effort devoted to reviewing our paper. As the discussion period is coming to an end, we are eager to know whether our response has addressed your concerns.
> >
> > We sincerely look forward to your feedback.
> >
> > Best regards,
> >
> > Authors

---

### Official Review · Reviewer_gVtC · 2024-10-21

**Soundness:** 3
**Presentation:** 2
**Contribution:** 2
**Rating:** 8
**Confidence:** 4

**Summary:**

To me this paper addresses the retraining problem in PINNs for solving PDEs. The authors introduce three methods: a basis solution approach for linear PDEs with variable conditions, a polynomial model for variable parameters, and a scaling method for certain PDEs. These techniques establish direct connections between PDE solutions and their configurations, enabling real-time inference without retraining. Theoretical analysis and extensive experiments demonstrate that these methods achieve comparable or better accuracy than traditional PINNs, while being significantly faster (often less than half-second inference time) and more effective in inverse problems. I like how this work makes PINNs more practical for real-time applications in scientific computing and engineering design.

**Strengths:**

1 The paper presents mathematically sound methods to directly connect PDE solutions with boundary conditions, sources, and parameters, addressing a key limitation of existing PINNs.
2.Theoretical foundation: The authors provide rigorous mathematical analysis and proofs for their proposed methods, including error bounds.
3.Computational efficiency: The methods achieve real-time inference (less than half a second in most cases) for both forward and inverse problems, significantly outperforming vanilla PINNs and meta-learning approaches.
4.Accuracy: The proposed methods demonstrate comparable or superior accuracy to vanilla PINNs in forward problems and significantly better performance in inverse problems.
5.Versatility: The approach is applied to various types of PDEs, including linear and nonlinear equations and includes extensive comparisons with state-of-the-art methods across multiple PDE benchmarks.

**Weaknesses:**

1. Limited scope: I feel like that the methods are primarily demonstrated on relatively simple PDEs and domains, although this is understandable.
2.Dependency on analytical insights: The approach relies heavily on analytical understanding of the PDEs, which may limit its applicability to more complex systems where such insights are not readily available.
3.Scalability concerns: It's unclear how well the methods would scale to very high-dimensional problems or systems of coupled PDEs. I feel like in general PINN's scalability is an issue.
4.Limited exploration of nonlinear PDEs: While the Reaction equation is considered, the treatment of more general nonlinear PDEs is not extensively explored.

**Questions:**

1. What are the limitations of your methods for highly nonlinear PDEs?
2. How does the performance degrade as PDE complexity increases? Is there a threshold where meta-learning approaches become more effective?
3. How does your approach extend to PDEs with complex geometries or higher dimensions?

---

> ### Author Response · Authors · 2024-11-21
> **Response to reviewer gVtC**
>
> Thank you very much for your insightful comments and valuable suggestions.
>
> 1.
> Q: only demonstrated on relatively simple domains.
>
> A: Our methods can be used for complex domains, despite we describe them using equations with simple domains. Please see the answer to question 1 in our responses to common questions for the explanation.
>
> 2.
> Q: Dependency on analytical insights.
>
> A: Although we propose our methods based on analytical insights on some equations, the resulted basis solution method and polynomial model are general to a large degree. Please see the answer to question 3 in our responses to common questions for the explanation.
>
> 3.
> Q: high-dimensional problems or systems of coupled PDEs.
>
> A: Our methods work for high-dimensional problems. Please see the answer to question 1 in our responses to common questions for the explanation.
>
> Our basis solution method applies to systems of coupled PDEs by solving PINNs for systems of coupled PDEs to obtain each basis solution. Our polynomial model also applies to systems of coupled PDEs with a single parameter, by modeling the solutions, e.g, as $u(x,t)=\displaystyle\sum\_{j=0}\^{N\_1}{w}\_{j}(x,t)(\beta/P)\^{j}$ and $v(x,t)=\displaystyle\sum\_{j=0}\^{N\_2}{w\^\prime}\_{j}(x,t)(\beta/P)\^{j}$.
>
> 4.
> Q: nonlinear PDEs.
>
> A: Our methods can be extended to non-linear equations. We have added new experimental results for the Burgers' equation. Please see the answer to question 2 in our responses to common questions for the detailed description.
>
>
> 5.
> Q: How does the performance degrade as PDE complexity increases? Is there a threshold where meta-learning approaches become more effective?
>
> A: Currently, vanilla PINNs are difficult to optimize for large parameter values (Krishnapriyan et al. NeurIPS 2021). As a result, the performance of our Fourier basis solution method will degrade as parameter gets large since we rely on vanilla PINNs to obtain basis solutions. Our polynomial model performs better than vanilla PINNs and meta-learning PINNs since the analytic relationship between solutions and parameters supports the generalization to big parameter values. This is also shown in our new experimental results for parameters in (0, 20], given in table 3 in the updated manuscript.
>
> Our methods will always perform better than or at least equal to meta-learning approaches. This is because in principle our basis solution method generalizes to arbitrary boundary/initial conditions and our polynomial model generalizes to arbitrary parameter values (subject to the accuracy limit of PINNs used to train basis solutions and coefficient functions) due to the established analytic relationship. Meta-learning approaches require multiple configurations as training tasks and the generalization performance depends on the training tasks used.
>
> 6.
> Q: How does your approach extend to PDEs with complex geometries or higher dimensions?
>
> A: Our methods are applicable to general domain geometry and high-dimensional problems. Please see the answer to question 1 in our responses to common questions for the detailed explanation.
>
> Thank you again for your insightful comments and constructive suggestions which help us to improve our manuscript.
>
>
> [1] Aditi Krishnapriyan, Amir Gholami, Shandian Zhe, Robert Kirby, and Michael W Mahoney. Characterizing possible failure modes in physics-informed neural networks. Advances in Neural Information Processing, 2021.

---

> > ### Comment · Reviewer_gVtC · 2024-11-21
> > **Response to reviewers**
> >
> > Thanks for the answers to my questions! Now it makes sense how it can be generalized to higher dimensions: )! The updated revision also now has more insights and its cool work! I have updated my score: )

---

> > > ### Author Response · Authors · 2024-11-21
> > > **Thank you**
> > >
> > > We sincerely thank you for your further comments and updating your score!

---

### Official Review · Reviewer_MsnK · 2024-11-04

**Soundness:** 3
**Presentation:** 3
**Contribution:** 2
**Rating:** 5
**Confidence:** 3

**Summary:**

This paper introduces two methods, Fourier series decomposition and the scaling method for PINNs to solve linear PDEs and one extra method, polynomial model, to solve potentially nonlinear PDEs. The Fourier series decomposition method pretrains sufficiently many basis PINNs which approximate the solution of the given linear PDE with initial condition being a Fourier basis function (sin, cos with given frequencies), so that when a new initial condition comes with the same PDE, they only need to apply Fourier series decomposition and linearly combine the corresponding PINNs’ solution with the Fourier coefficients. The scaling method apply change of variables to convection equation to reduce the coefficient $\beta$ to 1 (the canonical convection equation) so that the convection coefficient is small enough for MLP to learn the solution, then the method scales back the PINN’s solution to get the original solution. The polynomial model turns the PDE (linearity does not matter) into a polynomial regression problem and use neural network to predict coefficients of the polynomial.

**Strengths:**

- The paper is overall written clearly.

- The motivation is clear; the extension of PINNs into parameterized PDEs has yet not been studied extensively.

- The proposed method in some sense inherits the idea of reduced-order modeling (e.g., reduced-basis method), which is known to be a very-well established method.

**Weaknesses:**

- the Fourier series decomposition is a direct application of linearity of the three linear PDEs, and it is not expected that such method works for nonlinear equations (Burgers, etc.), so the application of this method can be still restricted. In 2D spatial domain, Fourier basis works only for rectangular domains with periodic boundary condition, leaving Poisson equation on other geometries of boundary (triangle, circle) and other types of boundary condition (Dirichlet, Neumann, etc.) unsolvable, while vanilla PINN works fine since it doesn’t need the Fourier basis on these domains. Going to a more general domain geometry or going to 3D domain could be very challenging.

- Polynomial method seems to be also restrictive. It is unclear how the method would provide prediction quality in extrapolation in the parameter space. Also, the number of polynomials seems to be fixed, which could be a strong modeling assumption. Lastly, did the authors consider different types of polynomials such as orthogonal polynomials? and how to handle the high-dimensionality?


- The scaling method (rebranding the change of variable) sounds a bit ad-hoc, which is designed for one class of equation only excluding Poisson). This method cannot be generalized to combined types of linear PDE, such as diffusion-reaction or diffusion-convection-reaction equations, which are the typical cases in PINN applications.


- Numerical experiments: Continuing from the previous comment, the numerical experiments do not include the cases with PDE equations such as diffusion-reaction or diffusion-convection-reaction equations.

- The numerical experiments are performed on rather a simple side, for example, the convection equation with $beta <=10$. The paper could have more impact if the method is employed on a rather hard side of the settings.

- As mentioned above in the polynomial method, it is unclear if the paper has performed tests regarding extrapolation in the parameter space / IC/ BC.

- Some ablation studies would be needed: The effect of number of Fourier bases, training and test setup (e.g., the number of different conditions to build bases), the number/type of polynomials, etc.

- Missing references: There have been some papers tackling issues in solving parameterized PDEs with PINNs. Some comparisons and/or discussions are needed.

  - Cho et al, Hypernetwork-based meta-learning for low-rank physics-informed neural networks, NeurIPS, 2023

  - Cho et al, Parameterized Physics-informed Neural Networks for Parameterized PDEs, ICML, 2024

**Questions:**

Please refer to the weaknesses for the questions.

---

> ### Author Response · Authors · 2024-11-21
> **Response to reviewer MsnK**
>
> Thank you very much for your insightful comments and valuable suggestions.
>
> 1.
> Q:  It is not expected that such method works for nonlinear equations (Burgers, etc.). In 2D spatial domain, Fourier basis works only for rectangular domains with periodic boundary condition, leaving Poisson equation on other geometries of boundary (triangle, circle) and other types of boundary condition (Dirichlet, Neumann, etc.) unsolvable. Going to a more general domain geometry or going to 3D domain could be very challenging.
>
> A: Our basis solution method works for general domain geometry,  other types of boundary condition (Dirichlet, Neumann. Not necessarily periodic) and high-dimensional problems, despite we take the Convection and Poisson equations etc. with simple rectangular domains and possible periodic boundary condition as concrete examples to describe our method. Please see the answer to question 1 in our responses to common questions for the detailed explanation.
>
> For nonlinear equations, we use our polynomial method to model the connections between solutions and boundary conditions. The dealing of nonlinear Reaction equation using polynomial method has been given in Appendix J. We have presented new experimental results on the Burgers' equation with varying parameters in table 3, table 4 and figure 5 in the updated manuscript. The polynomial model for the Burgers' equation with varying initial conditions is given in the answer to question 2 in our responses to common questions.
>
> 2.
> Q: It is unclear how the Polynomial method would provide prediction quality in extrapolation in the parameter space. Also, the number of polynomials seems to be fixed, which could be a strong modeling assumption. Lastly, did the authors consider different types of polynomials such as orthogonal polynomials? and how to handle the high-dimensionality?
>
> A: The polynomial method is already for extrapolation, which models the analytic relationship between solutions and parameters and does not require specific values of parameters to train the networks for the Convection and Heat equations. The results of polynomial method in table 3 are testing errors. **We further test the performance of the polynomial method for parameter values up to 20.
> The results for the Convection and Heat equations are given in table 3 and show that our polynomial method achieves much lower errors than vanilla PINNs which encounter optimization difficulties for large parameter values (Krishnapriyan et al. (2021)).**  **We also compare with** P$\^2$INNs (Cho et al. ICML 2024), and find that our method achieves lower errors, attributing to the explicit analytic connection between solutions and parameters in our model.
>
> **The number of polynomials $N\_p$ is set based on our theoretical analysis in Theorem 1 and Theorem 2**. More specifically, we set $N\_p=29$  to make $\frac{P\^{N\_p+1}}{N\_p!} < 1$ and consequently control the loss bound  $ (\max\_{x} \frac{\partial\^{N\_p+1} g(x)}{\partial x\^{N\_p+1}} )\^2 (\frac{P\^{N\_p+1}}{N\_p!}(\frac{\beta}{P})\^{N\_p+1})\^2 $.
>
> Thank you very much for your suggestion on orthogonal polynomials. Yes, we did thought of other possible approximation bases after deriving the polynomial model. We chose to use polynomials due to the following reasons: the polynomial model is inspired from the finite difference computation, hence it has enough expressive power to model the relationship between solutions and parameters, and it is the simplest one as compared with other models. It is interesting to explore other approximation bases in our future work.
>
> Our polynomial model $u(x,t)=\displaystyle\sum\_{j=0}\^{N\_p}{w}\_{j}(x,t)(\beta/P)\^{j}$ works in high-dimensionality. The optimization of ${w}\_{j}(x,t)$ is similar to that of $u(x,t)$ in vanilla PINNs, using residual loss and initial/boundary condition loss for ${w}\_{j}(x,t)$. Therefore, like vanilla PINNs, the polynomial model works in high-dimensionality through sampling collocation points.
>
> 3.
> Q: the numerical experiments do not include the cases with PDE equations such as diffusion-reaction or diffusion-convection-reaction equations.
>
> A: Thank you very much for your suggestions. Our current polynomial model applies to equations with one varying parameter. Diffusion-reaction and diffusion-convection-reaction equations combine several types of linear PDEs and thus involve multiple parameters in a single equation. Our current polynomial model can be applied to these equations when only one parameter is allowed to vary. We plan to explore more efficient models for multiple varying parameters in our future work.

---

> > ### Author Response · Authors · 2024-11-21
> > **Response to reviewer MsnK-continued**
> >
> > 4.
> > Q: The numerical experiments are performed on rather a simple side, for example, the convection equation with $\beta \le 10$. The paper could have more impact if the method is employed on a rather hard side of the settings.
> >
> > A: Thank you very much for your suggestions. We further test the performance of the polynomial method for parameter values up to 20 and compare with P$\^2$INNs (Cho et al. ICML 2024). Please see the answer to question 2 above for the details.
> >
> > 5.
> > Q:  it is unclear if the paper has performed tests regarding extrapolation in the parameter space / IC/ BC.
> >
> > A: We performed extrapolation in the parameter space in our polynomial method, please see our answer to question 2 above.  The extrapolation in IC/ BC has been fulfilled by our basis solution method and we have used significantly new ICs/sources as testing samples to obtain the extrapolation results in Appendix D.1 in our manuscript.
> >
> > 6.
> > Q: Some ablation studies would be needed: The effect of number of Fourier bases, training and test setup (e.g., the number of different conditions to build bases), the number/type of polynomials, etc.
> >
> > A: Thank you very much for your suggestions. For our Fourier basis method, the number of Fourier bases is originally set to ten to use only lower frequency bases, since as is well-known in signal processing, the boundary/initial values primarily consist of low frequency components. **We tried with more Fourier bases, including 15 and 20 bases, and the results are given in Appendix D.2.1 in the updated manuscript, which show that the errors almost remain constant and hence ten bases suffice to achieve accurate solutions.** In our Fourier basis method, there is no need to use conditions other than sine and cosine functions to build bases.
> >
> > The number of polynomials is determined theoretically, please see our answer to question 2 above.
> >
> > 7.
> > Q: Missing references: Some comparisons and/or discussions are needed.
> >
> > A: Thank you very much for your suggestions. The methods in these references have superior extrapolation performance. We have cited these papers in the updated manuscript and an empirical comparison with P$\^2$INNs (Cho et al. ICML 2024) is performed using the same range of parameter settings. The result is given in table 3 in the updated manuscript.
> >
> > Thank you again for your insightful comments and constructive suggestions which help us to improve our manuscript.
> >
> >
> > [1] Aditi Krishnapriyan, Amir Gholami, Shandian Zhe, Robert Kirby, and Michael W Mahoney. Characterizing possible failure modes in physics-informed neural networks. Advances in Neural Information Processing, 2021.
> >
> > [2] Woojin Cho, Minju Jo, Haksoo Lim, Kookjin Lee, Dongeun Lee, Sanghyun Hong, and Noseong Park. Parameterized physics-informed neural networks for parameterized pdes. In International Conference on Machine Learning, 2024.

---

> > > ### Comment · Reviewer_MsnK · 2024-11-25
> > >
> > > Thank you very much for the responses and the additional experiments. Thank you for the new descriptions on the general geometry and other types of boundary conditions, which could be an interesting direction that this method could be further investigated with some experimentations.
> > >
> > > However, seeing the experimental results and the response does not seem to address the concerns on generalizability or applicability of the method to broader problem classes. The new results on Burgers' equation seems to suggest that the method is struggling to achieve better accuracy than the vanilla PINN models, which may raise some doubts on if the method could achieve comparable performance on other problems (e.g., Navier--Stokes equations as the authors mentioned). Also, the fact that the method could be only applicable to a single scalar parameter case sounds like another major bottleneck. This is not to say that single scalar parametric scenarios are not important, but it is less convincing how effective the method is based on the presentation given so far. It might work with some more tuning (hyper-parameters, optimizer, etc), but it does seem that more time is needed to collect those results. Also, to show the effectiveness, the authors may want to consider more advanced problems (e.g., 1D Euler equation). Overall, the set of experimentation shown in this paper is on a weaker side and stronger empirical evidences would be needed.
> > >
> > > Regarding the arbitrary geometry: Could the authors expand their discussion on if and how Lemma 3 and the following would be modified if the problems are defined on other geometries? as Lemma 3 seems to assume the rectangular domain.
> > >
> > > Regarding the extrapolation:
> > > Also, the reviewer still has trouble in clear seeing the experimental setups. In the parameteric extrapolation, it can be guessed that the authors choose some parameter values (e.g., $\beta$) from a range (e.g,. [0,10]) during training and extend this range in the test case (e.g., [0, 20]). But it is still less certain on what exact $\beta$ values are chosen in the training phase and what exact values of $\beta$ are chosen in the test (and how they perform, and would there be some trend, e.g., the performance degrades as $\beta$ becomes larger)?
> > >
> > > Lastly, for IC/BC, the authors mention that they have chosen significantly different ICs/BCs for testing. Could the authors elaborate more on how the chosen ICs/BCs are significantly different from the ones of training? If the reviewer is missing, it would be nice if the authors could give a pointer.

---

> > > > ### Author Response · Authors · 2024-11-27
> > > > **Response to Official Comment by Reviewer MsnK**
> > > >
> > > > Thank you very much for your further feedback and insightful comments.
> > > >
> > > > Regarding the extrapolation: **We did not use any specific parameter values for the training of our polynomial model for the Convection, Heat and Reaction equations, this is where the insight and novelty come in. We achieve this through a series of innovations described by the mathematical formulations in Section 4.2.1**. We explain them in detail as follows. **Firstly**, we seek inspiration from the finite difference method which computes the solutions at interior points. Given a specific value of parameter $\beta$, in finite difference computation the solutions at interior points are actually expressed as a polynomial of $\beta$ (please refer to the expression given in lines 255-256 in our manuscript). Our key insight is that in this expression, the coefficients of the polynomials rely on locations but do not involve $\beta$, and the solutions depend on $\beta$ only in the form of $\beta\^j$. Therefore, we used the expression $u(x,t)= \sum\_{j=0}\^{N\_p}{w}\_{j}(x,t)(\beta/P)\^{j}$ to model the solutions (inspired by the stability analysis of finite difference method, $1/P$ is introduced to support convergence), where coefficient ${w}\_{j}(x,t)$ is a function of $x,t$ and does not rely on $\beta$. As a result, we can train ${w}\_{j}(x,t)$ offline and only once, after that the polynomial model can achieve fast inference for new values of $\beta$ in forward problems and quickly find the optimal value of $\beta$ in inverse problems. **Secondly**, in learning ${w}\_{j}(x,t)$, given the PDE for $u(x,t)$ (which is reduced to equation 8 in the manuscript) which in turn involves polynomials $\beta\^j$ with different order $j$, **our innovation here is to let all coefficients of $\beta\^j$ for any order $j$ (see equation 8) be equal to zero due to the fact that $\beta$ can have an arbitrary value, leading to a series of PDEs for ${w}\_{j}(x,t)$ (equations 9-11) that are totally irrelevant of $\beta$** (not only ${w}\_{j}(x,t)$ is irrelevant of $\beta$, but also the equations used to solve ${w}\_{j}(x,t)$ do not involve $\beta$). **Therefore, ${w}\_{j}(x,t)$ can be trained offline without using any specific values of $\beta$**. **Thirdly**, given the PDEs for ${w}\_{j}(x,t)$, they are solved just like in PINNs. We then **give a mathematical analysis for the loss bound of $u(x,t)$ brought by the solutions of ${w}\_{j}(x,t)$, and based on it, we give a theoretical criterion to determine the number of polynomials $N\_p$** for a given $P$ (the largest possible value of $\beta$).
> > > >
> > > > **In this process, we did not use any specific values of $\beta$ to train our model, which is inherently an extrapolation model by itself. This kind of model training is innovative and in stark contrast to the multi-task training manner used in meta-learning PINNs and P$\^2$INNs. This is one of our main contributions in this paper, enabled by the mathematical insights and rigorous analysis.**
> > > >
> > > > **When testing the extrapolation performance** of our polynomial model for parameter values up to 20, we set $P=20$ and $N\_p=60$ (which is enough to make $\frac{P\^{N\_p+1}}{N\_p!} < 1$). Given learned ${w}\_{j}(x,t)$ (irrelavant of $\beta$), **we compute the solutions $u(x,t)= \sum\_{j=0}\^{N\_p}{w}\_{j}(x,t)(\frac{\beta}{P})\^{j}$ directly for $\beta=1,2,3,\cdots,19$** and then compute the relative error for each of them.
> > > >
> > > > We found that when $\beta$ approaches 30 (to control the loss, $N\_p$ is set to 150), the arithmetic inaccuracy caused by computing $(\beta/P)\^{j}$ for large $j$ may accumulate, leading to inaccuracy in the solutions. Thanks for your insight. That is why we use learnable power instead of fixed one in our polynomial model for the Burgers' equation, shown in line 337 in the manuscript. We can still give PDEs for $w\_i (x,t)$ and $\phi\_{i} (x,t)$ that are irrelevant of parameter $\nu$, but currently it is hard to give a strict theoretical analysis on the loss bound of Burgers' equation, so we take the multi-task training scheme to train $w\_i (x,t)$ and $\phi\_{i} (x,t)$ in our current experiments. The parameter values of $\nu$ used for training are $0.03, 0.05, 0.07, 0.09, 0.1, 0.12, 0.14, 0.16$, respectively, and those used for testing are $0.01, 0.08, 0.15, 0.18, 0.20$, respectively, given table 13 in Appendix D.1.

---

> > > > > ### Author Response · Authors · 2024-11-27
> > > > > **Response to Official Comment by Reviewer MsnK-continued**
> > > > >
> > > > > **Regarding how the chosen ICs/BCs are significantly different from the ones of training:** In our basis solution method for varying initial/boundary condition problems, we use 20 bases $ cos(\frac{2\pi ix}{N}) $ and $sin(\frac{2\pi ix}{N})$, ($i=0,1,\cdots, 9$, $x=0,12,\cdots,N-1$ and $N=512$) as initial conditions to train the model. During testing, we use 6 initial conditions in the form of $sin(ax+b)$ ($x \in [0,2\pi]$ ) with different values of $a$ and $b$, e.g. $sin(3x+\frac{2\pi}{3})$, **whose phase is significantly different from that of base initial conditions used in training**. **These testing initial conditions are given in table 6 and table 7 in Appendix D.1.** **They are hard for the compared meta-learning models and PI-DeepONet to generalize.** **In contrast, our basis solution method accurately generalizes to arbitrary new initial conditions, enabled by the accurate reconstruction property of Fourier transform**. Figure 1 and figure 6 (in Appendix D.3) compare the effect of different methods on the testing initial condition $u(x,0)=\sin(3x+\frac{\pi}{3})$, which has **a phase shift** compared with training samples. The slices at t=0 and t=1 clearly show that our method can obtain much more accurate solutions (almost overlapping with the exact solutions),**while the unsuccessful generalization of other methods can be exhibited by the obvious shift of their solutions with respect to the exact ones.**
> > > > >
> > > > > **Regarding the arbitrary geometry for Poisson equation:** Our description in Lemma 3 and what follows is general, regardless of the domain geometry (though in our experiments a rectangular domain is used). If Poisson equation is defined on a domain with different geometry, then when solving basis solutions, we simply require them to satisfy the boundary conditions defined on the new boundary.
> > > > >
> > > > > **Regarding the applicability of our polynomial method to nonlinear equations and multi-parameter problems**: Our polynomial method is general and in principle applicable to nonlinear equations and multi-parameter problems. This is because our polynomial model is rewritten from the finite difference discretization of PDEs, thus **it inherits the generality of finite difference method and has enough expressive power to model the solutions for nonlinear equations and multi-parameter problems.** **For a PDE of general form $F(u(x,t),\mu) = f $**, where $F$ is the differential operator and $\mu$ is the parameter, we can always model the solution as $u(x,t) = \sum\_{i=0}\^{N} w\_i (x,t) \mu\^{\phi\_{i} (x,t)}$. For example, **for the Navier--Stokes equations** $u\_t+(u u\_x + v u\_y) = -\frac{1}{\rho} p\_x + \nu (u\_{xx}+u\_{yy})$ and $v\_t+(u v\_x + v v\_y) = -\frac{1}{\rho} p\_y + \nu (v\_{xx}+v\_{yy})$, where viscosity $\nu$ is the varying parameter, **we can model the solution as** $u(x,t)=\sum\_{j=0}\^{N\_1}{w}\_{j}(x,t) \nu\^{\phi\_{j} (x,t)}$ and $v(x,t)=\sum\_{j=0}\^{N\_2}{w\^\prime}\_{j}(x,t) \nu\^{\phi\^\prime\_{j} (x,t)}$. **For the diffusion-reaction equation** $u\_t - \nu u\_{xx} - \rho u (1-u)=0$ with two varying parameters $\nu$ and $\rho $, **we can model the solution as** $u(x,t)=\sum\_{j=0}\^{N\_1}{w}\_{j}(x,t) \nu\^{\phi\_{j} (x,t)} \rho\^{\psi\_{j} (x,t)}$. The derivations of these models are omitted here due to their lengthy discretized expressions. It is hard to include experiments for all these cases in a single paper in which we have performed experiments on five equations, including two nonlinear ones and both varying initial condition and varying parameter problems, so we leave the Navier-Stokes equation and the diffusion-reaction equation to our future work.
> > > > >
> > > > > **Regarding the results on Burgers' equation:** First we **have achieved better accuracy and much faster inference than vanilla PINNs in inverse problems (55 times faster)**, using only 250 samples of sensor data. Our method is **capable of dealing with inverse problems, this is one of main advantages of our method over other compared methods**. For the forward problem, we use vanilla PINNs to train $w\_i (x,t)$ and $\phi\_{i} (x,t)$, thus our error is subject to the error of vanilla PINNs. Our current result **for forward problems** is already promising, **being 170 times faster than vanilla PINNs** with a close relative error, demonstrating the superiority of our polynomial model. As you mentioned, the relative error can be further reduced through extensive fine-tuning.
> > > > >
> > > > > We sincerely thank you again for your insightful comments and look forward to your further response.

---

> > > > > > ### Comment · Reviewer_MsnK · 2024-12-02
> > > > > >
> > > > > > I'd like to thank the authors for providing more descriptions and the clarification. As the authors have addressed some concerns raised, the score has been updated accordingly.

---

> > > > > > > ### Author Response · Authors · 2024-12-02
> > > > > > > **Thank you**
> > > > > > >
> > > > > > > Thank you very much for your response and raising the score.

---

### Official Review · Reviewer_9Q83 · 2024-11-08

**Soundness:** 2
**Presentation:** 2
**Contribution:** 2
**Rating:** 5
**Confidence:** 4

**Summary:**

This paper addresses the limitations of Physics-Informed Neural Networks (PINNs), specifically the need for retraining when parameters or boundary/initial conditions of partial differential equations (PDEs) change. The authors propose a novel approach that leverages mathematical connections between PDE solutions and their boundary/initial conditions, sources, and parameters, enabling the model to solve new PDE configurations directly without retraining or fine-tuning. Experimental results show that this approach is up to 800 times faster than vanilla PINNs and 20 times faster than fine-tuning meta-learning PINNs, achieving inference times under half a second for forward problems and up to 3 seconds for inverse problems.

**Strengths:**

Unlike standard PINNs and meta-learning PINNs, this approach requires no retraining or fine-tuning to handle new PDE configurations, significantly reducing computational costs and simplifying model deployment. And the method achieves remarkable inference speed improvements, being up to 800 times faster than vanilla PINNs and around 20 times faster than meta-learning PINNs with fine-tuning. This efficiency makes it practical for real-time applications or situations where quick results are needed. Furthermore, by analyzing and leveraging mathematical structures in PDEs, the paper provides a principled method for establishing direct connections between PDE parameters and solutions. This reduces the reliance on purely data-driven methods and enhances interpretability.

**Weaknesses:**

**1. Parameter Complexity and Scalability**: The paper does not provide a detailed analysis of the total number of parameters required by the proposed method. For example, on cases involving linear PDEs with variable boundary/initial conditions or sources, since the model learns independent PINNs for each basis function and then combines them to approximate the solution, it appears that the parameter count could increase significantly. While the approach achieves adaptability to varying PDE configurations without fine-tuning, it may come at the cost of a substantial increase in model parameters, which could limit scalability and efficiency in more complex scenarios.

**2. Extension to Non-Linear and Complex PDEs**: The current method is demonstrated on relatively straightforward cases involving convection, heat, and reaction equations, which are linear PDEs in terms of their derivative terms. However, extending this approach to more complex, non-linear PDEs, such as the Burgers' or Navier-Stokes equations, would be challenging. These equations introduce additional complexities, such as non-linearity and turbulent behavior, which may not be easily captured by the current model’s parameterization and scaling methods. This limitation suggests potential difficulty in generalizing the method to more complex or chaotic systems, which restricts its broader applicability.

**Questions:**

Please refer to Weaknesses section.

---

> ### Author Response · Authors · 2024-11-21
> **Response to reviewer 9Q83**
>
> Thank you very much for your insightful comments and valuable suggestions.
>
> 1.
> Q: it may come at the cost of a substantial increase in model parameters, which could limit scalability and efficiency in more complex scenarios.
>
> A: We used in total twenty independent PINNs to train basis solutions for the Convection and Heat equations in our experiments. Each network used is a fully connected neural network of size [2, 100, 100, 100, 100, 1] and accordingly has a parameter count of approximately $3 \times 10\^4$.  Therefore, the parameter count is only $6 \times 10\^5$ for our method, which is very small compared with modern neural networks such as ResNets and Transformers while yielding 800 times acceleration over vanilla PINNs. Further, **inspired by your questions, we train a single network to produce all basis solutions and compare with training an independent network for each basis solution. These new results are given in table 15 in Appendix D.2.2 in the updated manuscript, which shows that training a single network can yield relative errors close to those of training independently, yet with smaller parameter count and faster training.** Thank you very much for your suggestions.
>
> Our basis solution method also scales to high-dimensions. Please see the answer to question 1 in our responses to common questions for the detail.
>
>  2.
> Q: extending this approach to more complex, non-linear PDEs, such as the Burgers' or Navier-Stokes equations, would be challenging.
>
>  A: Our method can be extended to non-linear equations. Please see the answer to question 2 in our responses to common questions for the detail. We have performed experiments on the Burgers' equation with varying parameters. The results show that our polynomial model has achieved lower error in inverse problems and is much faster than vanilla PINN, with a slightly higher error than it in forward problems.
>
> Thank you again for your insightful comments and constructive suggestions which help us to improve our manuscript.

---

> > ### Author Response · Authors · 2024-11-27
> > **Further response to reviewer 9Q83**
> >
> > Thank you very much for your insightful comments and valuable suggestions.
> >
> > Regarding the Burgers' equation, we **have achieved better accuracy (using only 250 samples of sensor data) and much faster inference than vanilla PINNs in inverse problems (55 times faster)**. Our method is **capable of dealing with inverse problems, one of main advantages of our method over other compared meta-learning methods**. **In forward problems, our method is 170 times faster than vanilla PINNs**, demonstrating the superiority of our polynomial model.
> >
> > Our polynomial model is rewritten from the finite difference discretization of PDEs, thus **it inherits the generality of finite difference method and has enough expressive power to model the solutions for nonlinear equations. For example, for the Navier--Stokes equations $u\_t+(u u\_x + v u\_y) = -\frac{1}{\rho} p\_x + \nu (u\_{xx}+u\_{yy})$ and $v\_t+(u v\_x + v v\_y) = -\frac{1}{\rho} p\_y + \nu (v\_{xx}+v\_{yy})$, where viscosity $\nu$ is the varying parameter,** **we can model the solution as $u(x,t)=\sum\_{j=0}\^{N\_1}{w}\_{j}(x,t) \nu\^{\phi\_{j} (x,t)}$ and $v(x,t)=\sum\_{j=0}\^{N\_2}{w\^\prime}\_{j}(x,t) \nu\^{\phi\^\prime\_{j} (x,t)}$.**
> >
> > We sincerely thank you again for your insightful comments , which have helped us greatly in improving our paper. We look forward to your response.

---

> > > ### Author Response · Authors · 2024-12-02
> > > **Looking forward to your feedback**
> > >
> > > Dear Reviewer 9Q83,
> > >
> > > Thank you very much for your insightful comments and constructive suggestions which have helped us greatly in improving our paper. As the deadline for the discussion is approaching, we would greatly appreciate it if you could let us know whether our responses have addressed your concerns. If you have any additional feedback, we would be most grateful to receive it.

---

### Author Response · Authors · 2024-11-21
**Responses to Common Questions**

Thank all reviewers very much for your hard work, insightful comments and valuable suggestions. The comments were helpful and constructive, enabling us to make significant improvements. The followings are our answers to the common questions raised by reviewers, including the generality of our methods, nonlinear equations, dependency on analytical insights and new experimental results. We have updated our manuscript accordingly and the modified parts are marked in red. We sincerely thank all reviewers again and look forward to further discussions.

1. **Our methods are applicable to general domains with complex geometry, high-dimensional problems and other types of boundary condition.**

A: **Our basis solution method** works for general domain geometry, other types of boundary condition and high-dimensional problems, despite we take the Convection and Poisson equations etc. with simple rectangular 2D domains and possible periodic boundary condition as concrete examples to describe our method. Following our description in the manuscript, the reason is explained as follows.

For the boundary of a  **domain (possibly high-dimensional) with arbitrary geometry**, the boundary values at every boundary point can be concatenated into a array $s(i)=g(\mathbf{x}\_i), \ \mathbf{x}\_i \in \mathrm {R}\^d \ \text{and} \ \mathbf{x}\_i \in \partial \Omega, \ i=0,1,2, \cdots, N-1 $, and then decomposed with one-dimensional FFT as $s(i) = \sum\^{N/2}\_{k=0} a\_k cos(\frac{2\pi ki}{N}) + b\_k sin(\frac{2\pi ki}{N}) $ (see Lemma 1). The one-dimensional bases $ cos(\frac{2\pi ki}{N}) $ and $sin(\frac{2\pi ki}{N}) $ can be inverse mapped into boundary points using $g\^{cos}\_k(\mathbf{x}\_i):=cos(\frac{2\pi ki}{N})$ and $g\^{sin}\_k (\mathbf{x}\_i):=sin(\frac{2\pi ki}{N})$,  $ i=0,1,2, \cdots, N-1$, respectively, and serve as boundary conditions for the high-dimensional and general domains. Basis solutions $u\^{cos}\_k(\mathbf{x},t)$ and $u\^{sin}\_k(\mathbf{x},t)$ are then obtained by training PINNs with boundary conditions $g\^{cos}\_k(\mathbf{x})$ and $g\^{sin}\_k (\mathbf{x})$, respectively, for several low frequencies $k$. Given an arbitrary boundary condition  $g(\mathbf{x})$, the corresponding solution is then obtained by the linear combination of basis solutions as $u(\mathbf{x},t) = \sum\^{N/2}\_{i=0} a\_i u\^{cos}\_i(\mathbf{x},t) + b\_i u\^{sin}\_i(\mathbf{x},t) $. Such $u(\mathbf{x},t)$ satisfies the linear equations under consideration,  and by $u(\mathbf{x},t)=\sum\^{N/2}\_{i=0} a\_i g\^{cos}\_i(\mathbf{x}) + b\_i g\^{cos}\_i(\mathbf{x}) =g(\mathbf{x}), \ \mathbf{x} \in \partial \Omega $, it also satisfies **the Dirichlet type of boundary condition**.

For the **Neumann type of boundary conditions** $ \frac{\partial u}{\partial \mathbf{n}}=g(\mathbf{x}), \ \mathbf{x} \in \partial \Omega $, we first convert $ g(\mathbf{x}\_i), \ \mathbf{x}\_i \in \partial \Omega $ into a array as above, and then train basis solutions $u\^{cos}\_k(\mathbf{x},t)$ and $u\^{sin}\_k(\mathbf{x},t)$ with Neumann type of boundary conditions: $\frac{\partial u\^{cos}\_k}{\partial \mathbf{n}} (\mathbf{x}\_i,t)=cos(\frac{2\pi ki}{N})$ and $\frac{\partial u\^{sin}\_k}{\partial \mathbf{n}} (\mathbf{x}\_i,t)=sin(\frac{2\pi ki}{N}), \ \mathbf{x}\_i \in \partial \Omega$, respectively. By $\frac{\partial u}{\partial \mathbf{n}} (\mathbf{x}\_i,t)=\sum\^{N/2}\_{i=0} a\_i \frac{\partial u\^{cos}\_i}{\partial \mathbf{n}} (\mathbf{x}\_i,t) + b\_i \frac{\partial u\^{sin}\_i}{\partial \mathbf{n}} (\mathbf{x}\_i,t) = g(\mathbf{x}\_i), \ \mathbf{x}\_i \in \partial \Omega $, the Neumann type of boundary conditions is satisfied.
Therefore,  our basis solution method works for general domain geometry, other types of boundary condition (Dirichlet, Neumann) and high-dimensional problems.

**Our polynomial model** $u(\mathbf{x},t)=\displaystyle\sum\_{j=0}\^{N\_p}{w}\_{j}(\mathbf{x},t)(\beta/P)\^{j}$ also works for complex domains, high-dimensional problems and other types of boundary condition (Dirichlet, Neumann). The optimization of ${w}\_{j}(\mathbf{x},t)$ is similar to that of $u(\mathbf{x},t)$ in vanilla PINNs, using residual loss and boundary/initial condition loss for ${w}\_{j}(\mathbf{x},t)$. Therefore, like vanilla PINNs, the polynomial model works for complex domains and high-dimensionality by sampling collocation points. Our polynomial model also works for both Dirichlet and Neumann boundary conditions
by optimizing ${w}\_{j}(\mathbf{x},t)$ with one of them.

We have added related discussions in the updated manuscript to emphasize the generality of our methods. .

---

> ### Author Response · Authors · 2024-11-21
> **Responses to Common Questions-continued-1**
>
> 2. **Nonlinear equations.**
>
>  A: Our method can be extended to non-linear equations. **Take the Burgers' equation to show how to extend our polynomial method to deal with varying parameters and initial conditions.** For the Burgers' equation $u\_t+u u\_x- \nu u\_{xx}=0$, inspired by its finite difference discretization $u\^{n+1}\_{j} = - u\^{n}\_{j} (1+\frac{2 \tau \nu}{h\^2})  - \frac{\tau}{h} u\^{n}\_{j} u\^{n}\_{j} + \frac{\tau}{h} u\^{n}\_{j} u\^{n}\_{j+1} - \frac{\tau}{h\^2} \nu (u\^{n}\_{j+1}- u\^{n}\_{j-1})$,  where $\tau$ and $h$ are the time and spatial step size, respectively,  **we use polynomial expression $u(x,t) = \sum\_{i=0}\^{N\_p} w\_i (x,t) \nu\^{\phi\_{i} (x,t)}$ to model the varying parameter problem**. We train $w\_i (x,t)$ and $\phi\_{i} (x,t)$ in this model in a multi-task manner using multiple  values of $\nu$ with corresponding residual loss and initial condition  loss for $u(x,t)$. **Our new experimental results for the Burgers' equation with varying parameter are given in table 3, table 4 and figure 5 in the updated manuscript**, which show that our polynomial model has achieved lower error in inverse problems and is much faster than vanilla PINN in inference (since retraining is not needed), with a slightly higher error than it in forward problems.
>
>  **We can use the polynomial expression $u(x,t) = \sum\_{i=0}\^{N\_p} w\_i (x,t) \prod\_j {(u\^0\_j)}\^{\phi\_{ij} (x,t)} $ to model the varying initial condition problem.** The training of it and the Navier-Stokes equation are leaved to our future work.
>
> 3. **Dependency on analytical insights.**
>
> A: Although we propose our methods based on analytical insights on some equations, the resulted basis solution method and polynomial model are general to a large degree. The Fourier basis solution method applies to any linear PDEs with complex domains and arbitrary boundary/initial conditions, and works for high-dimensional problems. The polynomial model gives a general form of dependency between solutions and parameters and is applicable to both linear and nonlinear equations with complex domains, arbitrary boundary/initial conditions and high-dimensionality. Please see our answers to the questions 1 and 2 above for the detailed explanations.
>
> Our theoretical analysis for the Convection and Heat equations gives some analytical insights on the loss bound. Based on it, we give a estimation for the number of polynomials used. If such analytical analysis is unavailable for other equations, one can simply rely on experiments to set the number of polynomials.
>
> 4. **New experimental results: ablation study and comparison with other methods.**
>
> A: Besides **the new results for the Burgers' equation in table 3, table 4 and figure 5**, we also performed the following new experiments.
>
> For our basis solution method, the number of Fourier bases is originally set to 10 for the Convection and Heat equations to use only lower frequency bases, since as is well-known in signal processing, the boundary/initial values primarily consist of low frequency components. **We tried with more Fourier bases, including 15 and 20 bases**, and the results are given in table 14 in Appendix D.2.1 of the updated manuscript, which shows that the testing errors are almost the same for different number of bases. Therefore, ten bases suffice to achieve accurate solutions.
>
> We also **train a single network to produce all basis solutions and compare with the results of training an independent network for each basis solution**. The results are given in table 15 in Appendix D.2.2 of the updated manuscript, which shows that training a single network yields only slightly higher relative errors with smaller parameter count and faster training.
>
> We then **test the performance of the polynomial method for parameter values up to 20**. The results for the Convection and Heat equations are given in table 3 and show that our polynomial method achieves much lower errors than vanilla PINNs which encounter optimization difficulties for large parameter values (Krishnapriyan et al. (2021)). We also compare with P$\^2$INNs (Cho et al. ICML 2024), and find that our method achieves lower errors, attributing to the explicit analytic connection between solutions and parameters in our model.

---

> > ### Author Response · Authors · 2024-11-21
> > **Responses to Common Questions-continued-2**
> >
> > Finally, we perform experiments to **compare our methods with PI-DeepONet (Wang et al. 2021)**. The results are shown in table 1 and table 2 in the updated manuscript, which show that our Fourier basis solution method significantly outperforms PI-DeepONet when training with the same set of input initial conditions. This is due to the fact that our Fourier basis solution method accurately reconstructs arbitrary boundary/initial conditions using only a limited number of low frequency Fourier bases, while PI-DeepONet requires a large number of diverse training samples of boundary/initial conditions to generalize. Our method is also faster than PI-DeepONet in both training and inference. Furthermore, inverse problem is not supported by PI-DeepONet.
> >
> > [1] Aditi Krishnapriyan, Amir Gholami, Shandian Zhe, Robert Kirby, and Michael W Mahoney. Characterizing possible failure modes in physics-informed neural networks. Advances in Neural Information Processing, 2021.
> >
> > [2] Woojin Cho, Minju Jo, Haksoo Lim, Kookjin Lee, Dongeun Lee, Sanghyun Hong, and Noseong Park. Parameterized physics-informed neural networks for parameterized pdes. In International
> > Conference on Machine Learning, 2024.
> >
> > [3] Sifan Wang, Hanwen Wang, and Paris Perdikaris. Learning the solution operator of parametric partial differential equations with physics-informed deeponets. Science advances, 7(40), 2021.

---

### Author Response · Authors · 2024-11-30
**A summary of our contributions and responses**

Dear reviewers,

Thank you for your constructive and insightful comments which have greatly inspired us to improve our paper, and for your valuable feedback and dedication during the discussions. Based on your feedback, we summarize our contributions in this paper as follows to facilitate further discussions.

1.
We propose the basis solution method and the polynomial method, establishing analytic connections between ICs/parameters and solutions empowered by mathematical insights and rigorous analysis, to tackle the retraining issue of vanilla PINNs. The basis solution method deals with varying IC/BC problems and applies to linear PDEs, and the polynomial method mainly deals with varying parameter problems and applies to linear/nonlinear PDEs.

Our basis solution method accurately generalizes to arbitrary new initial conditions by harnessing the accurate reconstruction property of Fourier transform, and only needs 20 sine and cosine initial conditions to train the model offline. In contrast, it is hard for meta-learning PINNs and PI-DeepONet to generalize to significantly new initial conditions.

Our polynomial method does not need any specific values of parameters for the model training and directly generalizes to the variation of parameters. This is in stark contrast to the multi-task training in meta-learning PINNs that have difficulties in generalizing to significantly new parameter values. Theoretical analysis for loss bound and the number of polynomials is established.

2.
Our methods are capable of dealing with inverse problems, which are not considered in meta-learning PINNs and PI-DeepONet.

3.
Our methods are general and applicable to general domains with complex geometry, different types of boundary conditions and high-dimensional problems. Our polynomial model has enough expressive power to model the solutions of general PDEs, including complex nonlinear equations and multi-parameter equations.

4.
Our methods are accurate and very fast, shown with experiments on five linear and nonlinear equations. The inference time is usually less than 0.5s in forward problems, and at most 3s in inverse problems (less than 0.5s for varying IC/BC problems) on a 3090 GPU. They are hundreds of times (over 800 times for varying IC/BC) faster than vanilla PINNs, and dozens of times faster than meta-learning PINNs. In forward problems, our methods are comparable to vanilla PINNs in accuracy and much more accurate than meta-learning PINNs and PI-DeepONet, and in inverse problems, our methods are much more accurate than vanilla PINNs.

To the authors' best knowledge, this is the first time inverse problems are solved in only seconds in the context of PINNs, using as few as 100 samples of sensor data.

We deeply believe that our work in this paper, as reviewer gVtC said, "is a cool work!", not only because of its novelties empowered by mathematical insights, but also its great potential in impacting the practical applications of PINNs.

We sincerely thank you again for your time and effort dedicated to reviewing our paper.

---

### Author Response · Authors · 2024-12-04
**New experimental results on Burgers' equation with varying initial conditions**

Dear reviewers,

Thank you very much for your time and effort devoted to reviewing our paper.

We work hard these days and have obtained new results on the Burgers' equation with varying initial conditions. We use our polynomial model $u(x,t) = \sum\_{i=1}\^{N\_p} w\_i (x,t) \prod\_{j=1}\^{N\_g}{(u\^0\_j)}\^{\phi\_{ij} (x,t)} $ to obtain the solutions of Burgers' equation $u\_t+u u\_x- \nu u\_{xx}=0$ without retraining when changing the initial condition $u\^0\_j$. The results on testing relative errors are as follows, which show that our errors  are very close to those of  vanilla PINNs.


 IC           |           Polynomial (ours)            |        vanilla PINN

----

 $\frac{5}{2}sin(\pi x)$         |             0.0371                 |               0.0275

 $\frac{7}{2}sin(\pi x)$         |            0.0227    |     0.0259

$\frac{9}{2}sin(\pi x)$         |            0.0314   |   0.0337

   average   |            0.0304   |   0.0293

----
The parameter $\nu$ is 0.05, $N\_p=50$, $N\_g=100$.

---

### Meta-Review · Area_Chair_vyNf · 2024-12-20

**Metareview:**

The work proposes an extension of the PINNs methodology for incorporating new boundary/initial condition data by a series/polynomial expansion based on the structure of the PDE in question. Numerical results are shown on several linear PDEs and a new Burgers' equation example has been added.

**Additional Comments On Reviewer Discussion:**

While the method rigorously derived and the theoretical analysis adds a lot of value, I share the reviews' concerns that numerics are too simple. While the authors added Burgers, this is really not a hard test case as it can be solved very quickly with traditional methods. More complex PDEs in, at least 2-d space + time,  or more complicated domain geometries have to added for the work to be published. The current numerics simply do not make a compelling enough case for the effectiveness of the method.

---

### Decision · Program_Chairs · 2025-01-22

Reject